# Inter- and Intraxylary Phloem in Vascular Plants: A Review of Subtypes, Occurrences, and Development

**Kishore S. Rajput** [1,*] **, Kailas K. Kapadane** [2] **, Dhara G. Ramoliya** [1] **, Khyati D. Thacker** [1] **and Amit D. Gondaliya** [1]

[1] Department of Botany, Faculty of Science, The Maharaja Sayajirao University of Baroda, Vadodara 390002, India
[2] Department of Botany, Faculty of Science, Dadasaheb Raval SVS's Science College, Dondaicha 425408, India
* Correspondence: ks.rajput15@yahoo.com

**Abstract:** Phloem is one of the vital tissues of the vascular system that plays a crucial role in the conduction of photosynthates. In vascular plants, it develops external to the vascular cambium but in a small fraction of eudicots (formerly known as dicots), it occurs within (interxylary) and inside (intraxylary) the secondary xylem. Ontogenetically, it is classified as *Strychnos*, *Combretum*, *Azima*, and *Calycopteris* types. In all four cases, phloem islands remain enclosed within the secondary xylem but each has unique origins. Similarly, the deposition of the phloem at the pith margin is common in several plants. It develops from procambial derivatives or adjacent pith cells or by initiating an intraxylary phloem cambium. Functionally, this cambium can produce only phloem or both secondary xylem and phloem. In some instances, the deposition of the secondary xylem and phloem in the same direction has also been documented. Some experimental evidence is available on the role of phloem but is it applicable to inter- and intraxylary phloem? The presence of inter- and intraxylary phloem is attributed to a defence mechanism against insects or plants that show sudden and enormous flowering or it can correlate with high temperatures or an unconducive climate in a desert region where sieve tube elements have become nonfunctional due to high temperatures. The present review is an attempt to analyse the role of interxylary and intraxylary phloem.

**Keywords:** phloem; interxylary phloem; intraxylary phloem; medullary phloem; internal cambium; sieve tube elements; variant secondary phloem

## 1. Introduction

Phloem is a crucial and principal conducting tissue in vascular plants that has a major role in the translocation of photosynthates from the source to the sink. Besides photosynthates, it also exchanges several other compounds including secondary metabolites and signalling molecules. Phloem has evolved as a specialized tissue for an inter-organ exchange mechanism that allows plants to retort and adapt to environmental and developmental stimuli [1]. Anatomically, a typical eudicot phloem is composed of sieve elements, companion cells, associated parenchyma, and sclerenchyma cells. Based on the time of initiation and growth stage, it may be primary (i.e., it develops in the embryos or young seedlings from the procambium) or secondary (develops from the vascular cambium) in origin [2]. Unlike the secondary phloem, which has an axial and radial system, the primary phloem lacks a radial (rays) system [2]. Young sieve elements contain all cellular components like other living cells of plants but as they begin to differentiate, the nucleus and tonoplast are broken gradually and develop specialized areas called sieve plates (which may be simple or compound) on the terminal ends. Angiosperms' sieve elements also show the presence of P-protein and callose that accumulates around the sieve pores of conducting phloem elements. In contrast, the non-conducting sieve elements show a heavy accumulation of callose, whereas seedless vascular plants are characterized by the presence of sieve cells and albuminous cells instead of sieve tube elements and companion cells.

Sieve tube elements are distinguished from sieve cells by having more specialized sieve areas with large sieve pores on the terminal end walls, whereas in sieve cells, they are present on the lateral walls [3]. In certain groups of vascular plants (specifically *Isoetes* in Lycopodiopsida and *Psilotum* in Polypodiopsida), it is often hard to decide whether a particular cell is a sieve tube element or a sieve cell [3]. This variation in terminology is associated with the cytological variations, arrangement of sieve areas, and ontogeny of their development. During the primary growth stage, in most of the eudicot stems, the phloem is part of a vascular bundle arranged in the same plane with the xylem (i.e., conjoint and collateral), whereas roots possess protostele in which the phloem lies in different radii and alternates with star-like protoxylem arms. However, in some members, such as cucurbits, morning glory, and several others (particularly representatives of Myrtales), the phloem is also present on the inner margin of the protoxylem. Such phloem present on the periphery of the pith (i.e., on the inner margin of the protoxylem) is variously known as medullary/internal or intraxylary phloem [4–10]. In most vascular plants, the phloem develops external to the vascular cambium (regular position) and its production is regulated by the activity of the vascular cambium. The deposition of the secondary phloem does not occur throughout the year, but is rather controlled by intrinsic and extrinsic factors and shows seasonality [11,12] due to the influence of environmental factors on cambial activity [2,11–16].

Before being designated the term 'phloem', it was referred to as bast [17] since attention was mainly focused on the phloem fibres. The use of this term may be associated with the economic advantage of the phloem fibres. The discovery of sieve tube elements by Hartig [18] changed phloem terminology. Thereafter, a distinction was made between soft bast (thin-walled elements such as sieve tube elements, companion cells, and associated parenchyma) and hard bast of the phloem fibres. After the recognition of their function as conducting elements and mechanical significance, Hartig [18], recognized them as complex tissues (c.f., [17]). Thereafter, Von Mohl [19] contributed significantly to the knowledge of the function of sieve tube elements and associated parenchyma, which contributed to them being regarded as physiologically essential elements. In 1858, Nägeli [20] coined the collective term 'phloem' for all the tissues by reserving the term bast of the fibres. The term 'phloem' was proposed by Nägeli and is now accepted by several researchers. However, some of the contemporary researchers at that time used other terms such as *libre* (bast) and *criblé* (sieve tissue) and the collective term '*Sieb*' or '*Cribraltheil*' [17]. Wilhelm [21] for the first time recognized and defined companion cells by providing further details about the difference between phloem parenchyma and sieve tube elements by considering the companion cells the sister cells of sieve tubes. Further studies by Hartig [18], Wilhelm [21], De Bary [22], and Lecomte [23] contributed significantly to the characteristics of the phloem parenchyma, sieve tube elements, and companion cells and distinguished between cytoplasm and slime. De Bary [22] was the first to observe the absence of a nucleus in sieve tube elements. The absence of a nucleus and its disintegration during the developmental stages was also confirmed by several researchers [21,24–31]. Besides the nucleus, it also showed the absence of ribosomes, alterations in the endoplasmic reticulum, and the widening of the symplasmic connections between sieve tube elements to form the sieve pores [32,33]. With the increase in knowledge, researchers at a physiological anatomy school proposed a new term '*leptome*' to describe the thin-walled unlignified tissue of the phloem including sieve tube elements [34]. Most of the information available to us on phloem comes from studies on seed plants, whereas similar studies on lycophytes and non-vascular plants are relatively scarce. Although reported solely in vascular plants, phloem-like rudimentary cells that conduct photosynthates are also known to occur in non-vascular plants such as leptoides in bryophytes and trumpet cells in kelps [35,36].

It is well established that the phloem plays a crucial role in the transport of photosynthates, nutrients, defensive compounds, signalling molecules (such as mobile proteins and RNA), long-distance macromolecular trafficking, and phytohormones [1,37–41]. Experimental studies have also been conducted to investigate the role of the phloem in

the conduction of photosynthates using carbon isotope ($^{14}$C) in different members of the Solanaceae family [42–46]. However, there is an urgent need to investigate phloem other than that regularly positioned (i.e., interxylary, intraxylary, or sieve tube elements recorded in the xylem and phloem rays) that is deposited by the vascular cambium during the secondary growth stage. To address this topic, we looked at the information available in the literature on why the phloem is deposited in unusual positions and whether its position is associated with different functions. For instance, Turgeon and Oparka [47] studied the interfascicular phloem of Cucurbit (pumpkin) and reported that the sugar content of the interfascicular phloem was around 30-fold less than the requirement. A meticulous study by these researchers revealed that the regular phloem has a higher sugar content than the fascicular phloem, whereas the extra-fascicular phloem has a lower sugar content. There-fore, the question arises as to whether the position of the phloem decides the function of the sieve tube elements. At present, there is no experimental evidence for why the phloem is formed within (interxylary phloem) or inside the xylem, i.e., at the periphery of the pith (intraxylary phloem), even though the regular phloem is present. Does it perform the same function as the regular phloem? Though there are several technological advancements in analytical instruments in recent years, there are difficulties with the sampling techniques for obtaining samples from an exact location in an intact plant. Although considerable information is available on phloem functioning (i.e., uploading and downloading), there is much more to be explored for the phloem that occurs in an unusual position, i.e., variant secondary phloem. Moreover, information available on unusual phloem is dispersed in the literature and there is a need for a comprehensive compilation of this information. The present work aims to compile information on phloem in unusual positions, as well as their ontogenies and possible roles, based on our work using supporting data from the available literature.

## 2. List of Species Investigated and Sample Processing

Detailed information about the species investigated in the present study is provided in Table 1. Information regarding sample processing (*viz.* sample collection, fixation, sectioning, and preparation of permanent slides, maceration, and photography) is descried in the references cited in the Table 1. For primary growth, samples were collected from the shoot tip up to the 25th node, whereas for secondary growth in climbers, samples were 5–15 mm thick. For trees, samples were excised from 25–40 cm stems, except for samples of *Adansonia digitata*, which were excised from 30 cm deep inside the stem of a naturally felled tree.

**Table 1.** The list of species investigated in the present study. The methods of sample analysis are discussed in the references cited in the last column. A similar method is also used for those species where the term 'present study' is mentioned.

| Sr. No. | Name of the Taxon | Family | Habit | References for Further Information |
|---|---|---|---|---|
| 1 | *Adansonia digitata* L. | Malvaceae | Tree | [48] |
| 2 | *Argyreia elliptica* (Roth.) Choisy | Convolvulaceae | Woody climber | [49] |
| 3 | *Argyreia nervosa* (Burm f.) Bojer | Convolvulaceae | Woody climber | [50] |
| 4 | *Argyreia osyrensis* (Roth.) Choisy | Convolvulaceae | Woody climber | [51] |
| 5 | *Argyreia sericea* Dalzell and A. Gibson | Convolvulaceae | Woody climber | [51] |
| 6 | *Argyreia splendense* (Hornem) Sweet | Convolvulaceae | Woody climber | [51] |
| 7 | *Arthrocnemum indicum* (Willd.) Moq. | Amaranthaceae | Shrub | Present study |
| 8 | *Barleria prionitis* L. | Acanthaceae | Herb/shrub | Present study |
| 9 | *Beaumontia jerdoniana* Wight | Apocynaceae | Woody climber | [52] |

**Table 1.** *Cont.*

| Sr. No. | Name of the Taxon | Family | Habit | References for Further Information |
|---|---|---|---|---|
| 10 | *Boerhavia diffusa* L. | Nyctaginaceae | Diffuse herb | [53] |
| 11 | *Bougainvillea* sp. | Nyctaginaceae | Scandent shrub | Present study |
| 12 | *Bombax ceiba* L. | Malvaceae | Tree | [54] |
| 13 | *Calycopteris floribunda* (Roxb.) Lam. | Combretaceae | Woody lianas | [55] |
| 14 | *Campsis radicans* (L.) Bureau | Bignoniaceae | Woody climber | [56] |
| 15 | *Camonea vitaefolia* (Burm.f.) A.R Simoes and Staples | Convolvulaceae | Climber | Present study |
| 16 | *Canavalia ensiformis* (L.) DC. | Fabaceae | Annual climber | [57] |
| 17 | *Canavalia gladiata* (Jacq.) DC. | Fabaceae | Annual climber | Present study |
| 18 | *Coccinia grandis* (L.) Voigt | Cucurbitaceae | Climber | [58] |
| 19 | *Dalechampia coriacea* Klotzsch ex Müll.Arg. | Euphorbiaceae | Climber | Present study |
| 20 | *Dicranostylis ampla* Ducke | Convolvulaceae | Woody liana | [59] |
| 21 | *Diplopterys carvalhoi* W.R. Anderson and C.Davis | Malpighiaceae | Woody liana | Present study |
| 22 | *Distimake tuberosus* (L.) A.R. Simõesand Staples | Convolvulaceae | Woody liana | [60] |
| 23 | *Dolichandra unguis-cati* (L.) L.G. Lohmann | Bignoniaceae | Woody liana | Present study |
| 24 | *Erythrina indica* Lam. | Fabaceae | Tree | [54] |
| 25 | *Gallesia integrifolia* (Spreng.) Harms. | Phytolaccaceae | Tree | [61] |
| 26 | *Heteropterys* sp. | Malpighiaceae | Woody liana | Present study |
| 27 | *Hewittia malabarica* (L.) Suresh | Convolvulaceae | Woody climber | [62] |
| 28 | *Ipomea hederifolia* L. | Convolvulaceae | Annual climber | [63] |
| 29 | *Ipomoea muricata* (L.) Jacq. | Convolvulaceae | Climber | Present study |
| 30 | *Ipomoea pes-caprae* (L.) R.Br. | Convolvulaceae | Perennial climber | [64] |
| 31 | *Ipomoea turbinata* Lag. | Convolvulaceae | Annual climber | [65] |
| 32 | *Jacquemontia pentantha* (Jacq.) G.Don | Convolvulaceae | Perennial climber | Present study |
| 33 | *Leptadenia reticulata* (Retz.) Wight and Arn. | Apocynaceae (Asclepiadaceae) | Climber | [66] |
| 34 | *Leptadenia reticulata* (Retz.) Wight and Arn. and *L. pyrotechnica* (Forssk.) Decne. | Apocynaceae (Asclepiadaceae) | Climber and shrub respectively | [67] |
| 35 | *Leptadenia pyrotechnica* (Forssk.) Decne. | Apocynaceae (Asclepiadaceae) | Shrub | [68] |
| 36 | *Mansoa alliacea* (Lam.) A.H.Gentry (syn. *Bignonia alliacea*) | Bignoniaceae | Climber | Present study |
| 37 | *Merremia hederacea* (Burm.f.) Hallier f. | Convolvulaceae | Climber | Present study |
| 38 | *Mucuna pruriens* (L.) DC. | Fabaceae | Climber | Present study |
| 39 | *Neuropeltis racemosa* Wall. | Convolvulaceae | Woody liana | Present study |
| 40 | *Phaseolus lunatus* L. | Fabaceae | Annual climber | Present study |
| 41 | *Pupalia lappacea* (L.) Juss. | Amaranthaceae | Perennial shrub | [69] |
| 42 | *Salvadora persica* L. | Salvadoraceae | Tree | Present study |

**Table 1.** *Cont.*

| Sr. No. | Name of the Taxon | Family | Habit | References for Further Information |
|---|---|---|---|---|
| 43 | *Salvadora oleoides* Decne. | Salvadoraceae | Tree | Present study |
| 44 | *Solanum pseudocapsicum* L. | Solanaceae | Perennial shrub | [70] |
| 45 | *Serjania mexicana* (L.) Willd. | Sapindaceae | Woody climber | [71] |
| 46 | *Strychnos bicolor* Prog. | Loganiaceae | Straggling shrub | [72] |
| 47 | *Strychnos bredemeyeri* (Schult. and Schult. f.) Sprague and Sandwith | Loganiaceae | Straggling shrub | [73] |
| 48 | *Strychnos potatorum* L.f. | Loganiaceae | Tree | Present study |
| 49 | *Sterculia urens* Roxb. | Malvaceae (Sterculiaceae) | Tree | [74] |
| 50 | *Strychnos potatorum* L. | Loganiaceae | Tree | Unpublished data |
| 51 | *Tectona grandis* L.f. | Lamiaceae | Tree | [54] |
| 52 | *Turbina corymbosa* (L.) Raf. | Convolvulaceae | Perennial climber | [75] |
| 53 | *Thunbergia grandiflora* Roxb. | Acanthaceae | Climber | Present study |
| 54 | *Suaeda fruticosa* Forssk. ex J.F.Gmel. | Amaranthaceae | Small tree | [76] |
| 55 | *Suaeda nudiflora* Forssk. exJ.F.Gmel. | Amaranthaceae | Small tree | [76] |
| 56 | *Vallaris solanacea* (Roth) Kuntze | Apocynaceae | Woody climber | [67] |

## 3. Inter- and Intraxylary Phloem in Vascular Plants

During evolution, the origin of vascular tissues in the early land plants changed the vegetation pattern on the planet, which consequently resulted in the enormous phytodiversity of land plants. These vascular tissues are composed of more than one or two types of cells that faultlessly coordinate with each other to communicate and perform their roles accurately in the conduction of different kinds of solutes such as photosynthates, minerals, nutrients, various signalling molecules related to different functions, and molecules that are vital in defence mechanisms [33,39,42]. Owing to their composition of multiple cell types (i.e., xylem and phloem), they are also referred to as complex tissues. Both the xylem and phloem are necessary tissues that play a pivotal role in the survival of every species, even growing under harsh conditions. Investigating the chronology of the evolution/modifications and alterations in both the tissues (i.e., xylem and phloem) in various groups of plants (such as magnoliids, eudicots, monilophytes, and lycophytes) is interesting. Within angiosperms themselves, monocots differ from eudicots, and within eudicots, self-supporting and non-self-supporting plants show tremendous alterations in the architectures of these tissues and their compositions, structures, arrangements, locations, and the increases in their stem diameters [4,5,7,9,34,77–88]. During evolution, to compete for above- and underground resources [89,90], countless changes occur in the structure, composition, and location of the conducting tissues and the meristem responsible for their deposition. Some of these changes are furrowed/fissured xylem, multiple rings of xylem and phloem (successive rings), enclosing the phloem in the secondary xylem (interxylary phloem), phloem on the pith margin (intraxylary phloem), medullary bundles, cortical bundles, etc. All these features are most often referred to as variant secondary growth, which is directly or indirectly associated with the conduction of photosynthates and mechanical support [49,74,83–87]. To the best of our knowledge, the oldest known records of cambial variants are those by De Mirbel in 1828 [91] and subsequently, several reports [5,92–94] have appeared on this topic but even today, no concrete proof is available of their functionality, except for a few experimental studies [1,32,33,37–46].

All these alterations in the histological architecture of the plants are part of a process called "Survival of the Fittest", as expressed by Darwin [95]. All these unusual

secondary growth patterns have evolved multiple times [96–98] and independently in different plant families, and some of them are so characteristic of a certain group of plants that their families can be identified based on their stem characteristics [99,100]. For example, the Convolvulaceae family is characterised by the presence of intraxylary phloem and successive cambia [10,63,74], the Menispermaceae family is characterised by successive cambia [77,101], the Sapindaceae family is characterised by a compound or divided stem [77,102–104], and the tribe Bignonieae in the Bignoniaceae family is characterized by the presence of furrowed xylem and phloem wedges [22,77,100,105]. A combination of multiple types of cambial variants is also frequently observed in several members of these families. All these alterations in the vascular system of non-self-supporting species are associated with explicit physical constraints due to their narrow stems for the conduction of water, minerals, and nutrients [97,106]. The occurrence of these types of variant secondary growth is attributed to an increase in stem flexibility, which can twine around the supporting object [83,84,86]. However, several members are self-supporting yet possess multiple cambia or other types of cambial variants [61,107,108]. Besides the mechanical properties of stem flexibility, the formation of phloem by various means in positions other than the regular position is also documented in the literature, which is presented below.

### 3.1. Interxylary (Included) Phloem

Interxylary phloem (also referred to as "included phloem") are islands or strands of sieve elements, companion cells, associated parenchyma cells, or other types of cells (Figure 1A) embedded within the secondary xylem of the stem/roots of the magnoliids and eudicots that increase in their thickness by a single ring of the vascular cambium [109]. The occurrence of interxylary phloem has been reported in 21 families by Carlquist [9]. Further, he commented that some of them need further study. According to Carlquist, in the earlier literature, the term interxylary phloem was likewise used to describe the phloem produced by successive cambia as concentric rings. However, the phloem produced by the successive cambia is ontogenetically normal in a position [9]. Therefore, in the present study, the term interxylary phloem refers to the phloem that is a product of a single cambium. Although the members of the Convolvulaceae family show the presence of successive cambia, in the present study, the examples cited from this family include the portion of the xylem that is produced by the first regular vascular cambium. Based on their ontogeny, they are recognized as the *Combretum*, *Strychnos*, *Azima,* and *Calycopteris* subtypes [110,111]. In all four subtypes, a single ring of the vascular cambium remains functional throughout life. Although the subtypes are genus-specific, their pattern of development has also been reported in other genera of different families. It has been observed that even members of the same family show variations in development, for example, *Combretum* and *Calycopteris*. Although both belong to the same family, the development of interxylary phloem shows significant variations. However, there is no separate list of interxylary phloem based on the above-mentioned types and the information is simply based on the phloem enclosed within the secondary xylem that is a product of a single cambium.

Ontogenetically, interxylary phloem is the intricate process when members of the same species show multiple combinations of variant secondary growth. For instance, several members of the Convolvulaceae family possess successive cambia and inter- and intraxylary phloem. Structurally, the xylem formed by the regular (first) vascular cambium produces islands of unlignified parenchyma cells that dedifferentiate into interxylary phloem after a certain period. As per our observations, its development was observed when the plants reached the reproductive phase.

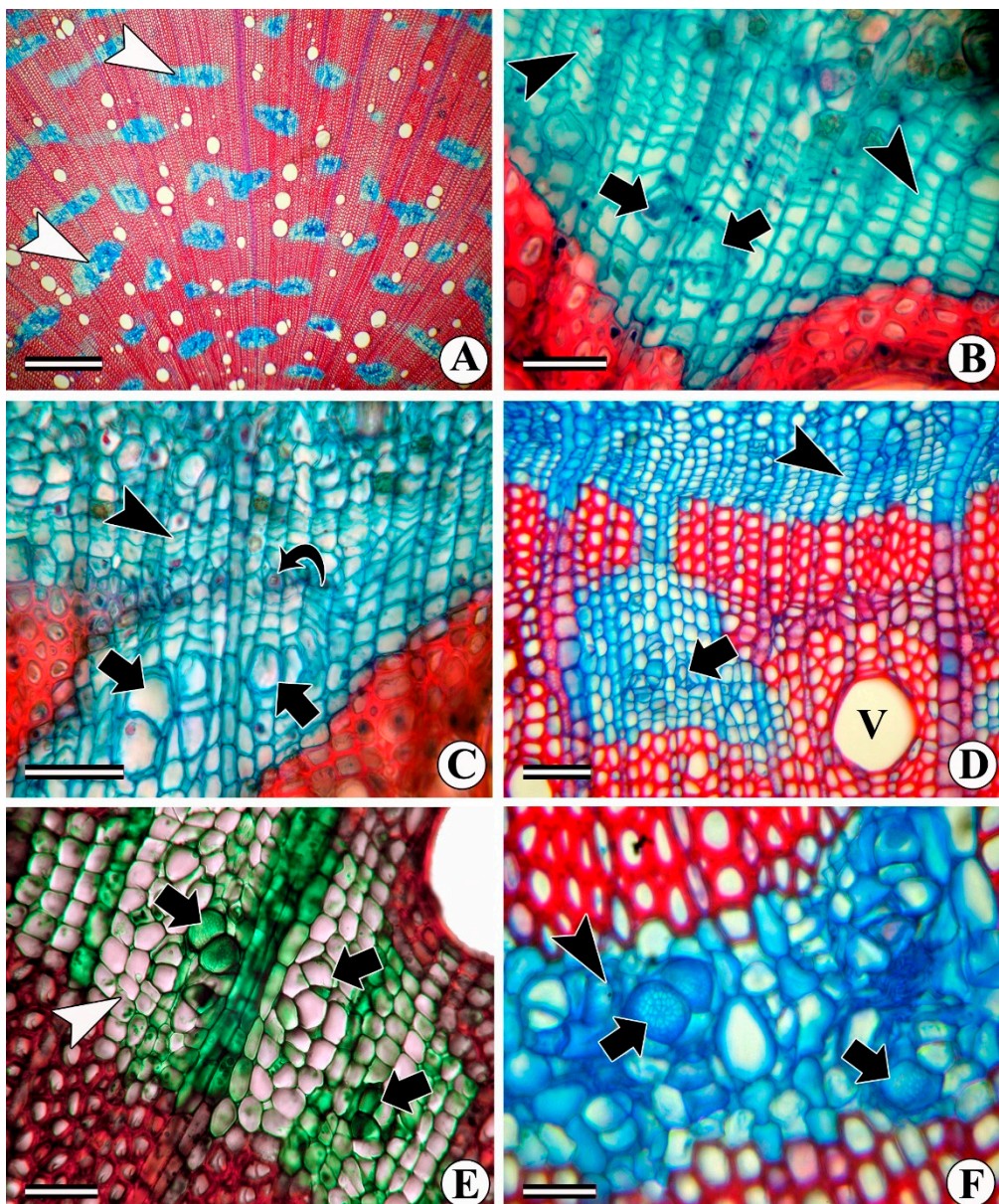

**Figure 1.** Transverse view of the secondary xylem showing various stages of interxylary phloem development in different species of eudicots. (**A**) Interxylary phloem islands (arrowheads) in the stem of *Leptadenia pyrotechnica*. (**B**) Differentiating sieve elements from the daughter cells (arrows) of the cambium (arrowheads) in the stem of *Mucuna pruriens*. (**C**) Differentiation of lignified elements (curved arrow) after a short period of phloem development in the stem of *M. pruriens*. Note the cambium (arrowhead) and centripetally developed sieve elements (arrows). (**D**) Interxylary phloem (arrow) embedded within the secondary xylem of *L. pyrotechnica* stem. Arrowhead indicates vascular cambium. *Abbreviation*: V = vessel (**E,F**) Relatively enlarged view of the sieve tube elements (arrows) enclosed within the xylem of *L. reticulata* (**E**) and *L. pyrotechnica* (**F**). Arrowheads indicate phloem-associated parenchyma. ***Scale bars***: (**A**) = 500 μm; (**B,C**) = 100 μm; (**D**) = 200 μm; (**E**) = 75 μm; (**F**) = 50 μm.

### 3.1.1. Combretum Subtype

In this category, secondary growth is initiated like other woody angiosperms that produce secondary xylem centripetally and secondary phloem centrifugally [112,113]. However, after the production of a few derivatives of the secondary xylem, small segments of the vascular cambium start the inward development of phloem elements instead of the xylem.

After a short period of inward phloem deposition, cambial activity resumes its regular activity and starts producing lignified xylem derivatives internally (Figure 1A–C). Similar behaviour of interxylary phloem has also been reported in *Mucuna* by Solereder [114]. Subsequently, several researchers have described interxylary phloem development in several other species [7,68,79,109,115,116]. Such activity of the cambium (i.e., inward phloem formation) operates repeatedly and independently throughout the cambial ring, which consequently encloses several phloem islands distributed randomly throughout the secondary xylem (Figure 1D). The ratio of sieve tube elements and associated parenchyma varies from species to species. In the case of *Barleria prionitis* and its other species, *Leptadenia reticulata*, *L. pyrotechnica*, *Thunbergia grandiflora,* etc., initially, every phloem island is mostly composed of associated parenchyma, whereas sieve tube elements account for a small portion (Figure 1C–E). As the secondary growth stage progresses, the associated parenchyma cells dedifferentiate and deposit additional sieve tube elements, and gradually, a complete parenchyma patch is occupied by sieve tube elements (Figure 1F). In some of the samples, these parenchyma cells become meristematic and show radial files of tangentially flattened meristematic cells appearing like cambium (Figure 2A,B). Besides the production of sieve elements, some of the cambial segments exclusively deposit parenchyma patches that lack sieve elements (i.e., form parenchyma patches enclosed within the secondary xylem). In time, these parenchyma cells dedifferentiate and lead to the formation of sieve elements (Figure 2C), which indicates the co-occurrence of multiple cambial variants within the same species. However, the formation of such interxylary phloem (i.e., delayed development) is best suited to the *Azima* subtype; thus, it is considered in a later section. In *Barleria prionitis* and *B. grandiflora*, phloem islands are small and possess only one or a few sieve elements and one to two associated parenchyma (Figure 2D). A sieve tube may have one to two companion cells (Figure 2E,F).

### 3.1.2. Strychnos Subtype

In this subtype, all the investigated species (irrespective of habit, i.e., shrubs, trees, or sarmentose lianas) follow a similar pattern of secondary growth. Initially, primary and secondary growth in all the species of *Strychnos* studied so far commence as in other eudicots. A single ring of the vascular cambium remains functional throughout its lifespan and maintains a circular outline of the stem. After a short period of regular secondary growth, the inward development of the secondary xylem from small segments of the vascular cambium retards gradually and consequently, ceases to produce the secondary xylem in several places. This leads to the formation of shallow invagination/arcs towards the xylem side; therefore, the adjacent cambial segments are detached and continuously pushed centrifugally due to regular cell division and the deposition of the xylem (Figure 3A). Subsequently, the phloem parenchyma that are external to the shallow invagination/arcs become meristematic and form coalescent cambial segments (Figure 3B) that are connected to the regularly functioning cambium and form a complete ring. These newly formed cambial segments produce secondary xylem centripetally and secondary phloem centrifugally. Consequently, these depressions containing phloem become embedded within the secondary xylem (Figure 3C). This process is repeated randomly in several places throughout the cambial ring and forms a foraminate type of interxylary phloem (Figure 3D). The segment of the cambium associated with every phloem island retains its radial arrangement, whereas non-conducting phloem elements are replaced by adding new sieve elements from the associated cambial segment (Figure 3E,F). The pressure exerted from the newly added sieve elements consequently crushes the non-conducting phloem and leads to the accumulation of collapsed phloem opposite the cambial segment (Figure 3D). This type of interxylary phloem develops due to the temporary cessation of cell division activity of a small portion of the cambium and the formation of a coalescent cambial segment externally to uphold a complete ring of the vascular cambium. This type of variant secondary growth was first reported by Fritz Muller [117] in *Dicella* of the Malpighiaceae family (c.f., [4]). Subsequently, De Bary [22] investigated this in detail using dry samples, and the ontogeny described is

still accepted today. Hérail [118] studied several variant secondary growth structures in other woody angiosperms including *Strychnos*. Thereafter, numerous reports appeared on a similar kind of interxylary phloem development [4,72,73,76,108,113,119]. De Bary [22] compared the development of interxylary phloem in *Strychnos* with some members of the Chenopodiaceae family.

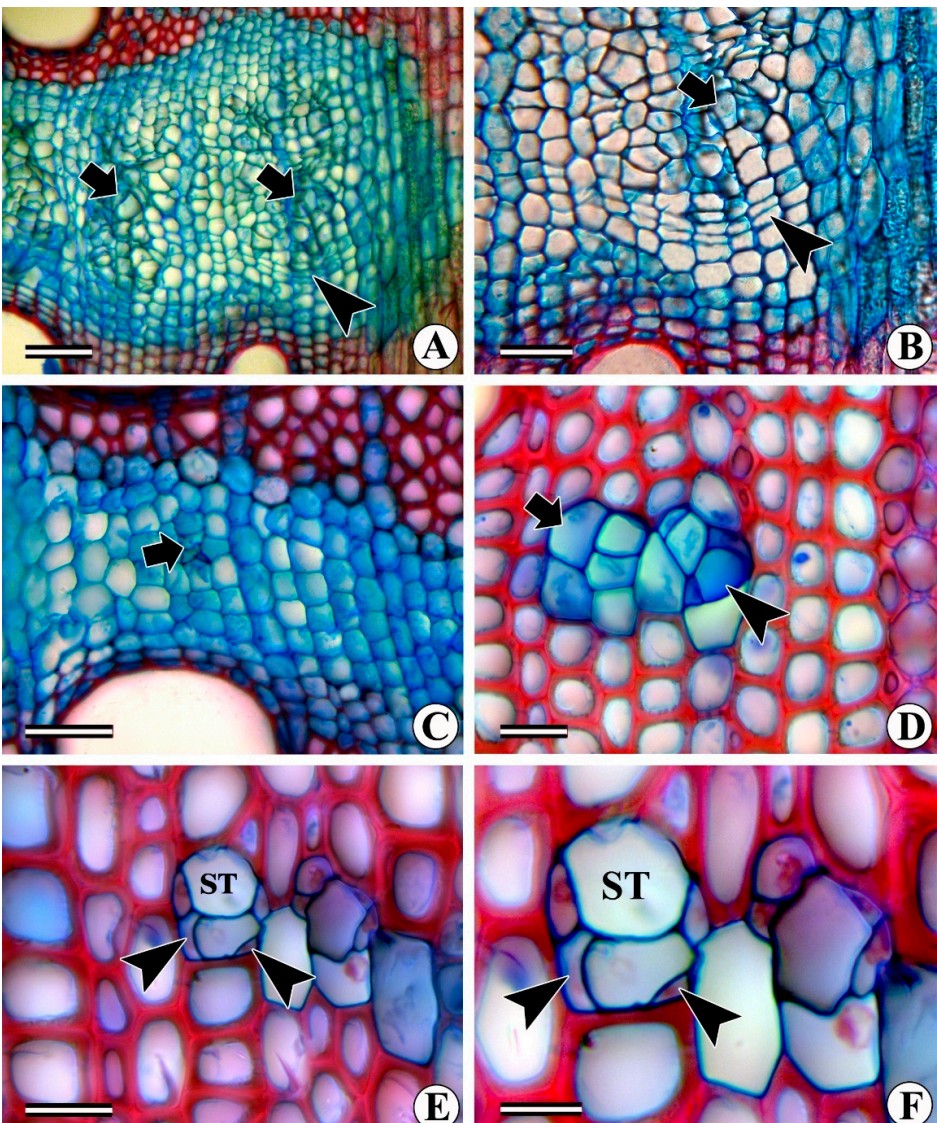

**Figure 2.** Transverse view of the secondary xylem showing interxylary phloem. (**A**) One of the phloem islands in *Leptadenia reticulata* showing the initiation of cell division in the associated parenchyma (arrowhead). Note the arrangement of cells in the radial file. Arrows indicate the sieve elements. (**B**) Enlarged view of Figure 2A showing the divisions in the associated parenchyma and the arrangement of dividing cells arranged in radial files like cambium (arrowhead). The arrow shows the sieve tube element. (**C**) One of the parenchyma islands in *L. reticulata* showing differentiating sieve elements (arrow) showing delayed development of phloem elements. (**D**) Secondary xylem of *Barleria prionitis* showing an interxylary phloem island composed of a few sieve elements (arrowhead) and associated parenchyma cells (arrow) (**E,F**) *Barleria* sp. ((**F**): enlarged view of Figure 2E) showing sieve elements (ST) with two companion cells (arrowheads). Note the companion cells with each sieve tube element. *Abbreviation*: ST = sieve tube element. **Scale bars**: (**A**) = 200 μm; (**B,C**) = 100 μm; (**D,E**) = 50 μm; (**F**) = 20 μm.

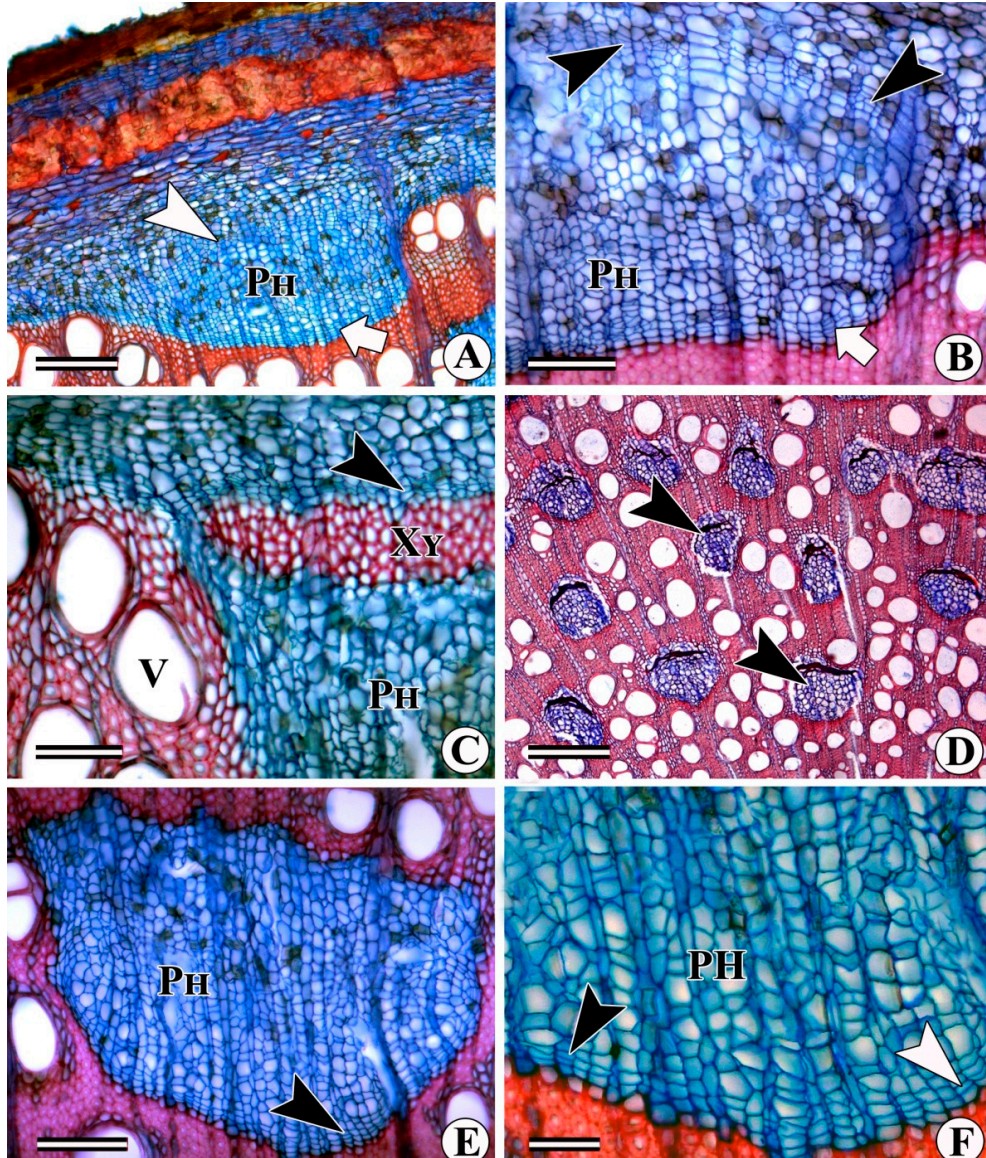

**Figure 3.** Transverse view of secondary xylem of *Strychnos bredemeyeri* (**A–C**), *S. bicolor* (**E**), and *Salvadora oleoides* (**F**) showing interxylary phloem strands. (**A**) Development of interxylary phloem island. Cessation of cambial cell division and formation of depression (arrow) followed by the development of a new cambial segment (arrowhead). Arrow indicates cambium in the depression. (**B**) Enlarged view of developing interxylary phloem island. Note the shallow depression and cambium (arrow). Arrowheads show a newly initiating cambial segment. (**C**) Initiation of deposition of lignified xylem derivatives (XY) by the cambium (arrowhead). (**D**) Interxylary phloem strands (arrowheads) embedded in the secondary xylem. (**E**) One of the interxylary phloem islands (PH) embedded within the secondary xylem. Note the cambium (arrowhead) on the inner margin of the phloem strand. (**F**) Enlarged view of Figure 3E showing radially arranged cambial cells (arrowheads) in one of the phloem islands. *Abbreviations*: PH = phloem embedded within the secondary xylem, V = vessel, XY = xylem. ***Scale bars***: (**A**,**D**) = 200 μm; (**B**,**C**,**E**) = 100 μm, (**F**) = 50 μm.

A similar kind of coalescent cambial segment formation can be observed during the early stages of secondary growth in several members of the Amaranthaceae (including Chenopodiaceae), Nyctaginaceae, and Phytolaccaceae families. However, as secondary growth progresses, the tangential width of coalescent cambia increases gradually and in thick stems, a complete ring of the cambium is renewed by the new cambial ring [76,120]. Recently, Cunha Neto et al. [121] reported two different types of cambial variants in

Nyctaginaceae, i.e., the formation of interxylary phloem and successive cambia. The authors of the present study agree with Cunha Neto et al. [121] because interxylary phloem islands are the product of coalescent cambia. According to Carlquist [122], the use of the term 'interxylary phloem' in the Caryophyllalean families, particularly the 'centrospermoide' families, is a misnomer since the phloem in successive cambia is embedded within the background (conjunctive) tissues, although it appears woody in texture. This may be true in thick stems where the cambia form a complete ring but in relatively young stems, only small segments of the cambia are renewed. Therefore, the phloem islands (not forming a complete ring) are not enclosed in the conjunctive tissue but the phloem is enclosed within the coalescent parenchyma.

### 3.1.3. Azima Subtype

This type of interxylary phloem development is a common phenomenon in several magnoliids and eudicots. In this subtype, certain parenchyma cells that formed earlier during vegetative growth, which were embedded within the secondary xylem, dedifferentiate and re-differentiate into sieve elements [123,124]. In the present study, this was commonly observed in *Coccinia grandis*, *Canavalia ensiformis*, *C. gladiata*, *Phaseolus lunatus*, *Salvadora oleoides* (Figure 4A,B), and some species of the Convolvulaceae family (*Hewittia malabarica*, *Ipomoea hederifolia,* and *I. turbinata*). Similar meristematic activity has also previously been documented in *Craterisiphon* by Carlquist [109] and showed the absence of phloem initially while their number increased with the age of the plant. Schenck [77] and Pfeiffer [78] also described the differentiation of sieve elements in the stems of *Mucuna altissima* DC. As per our observations (at least in the Convolvulaceae family), the differentiation of sieve elements in these parenchyma cells was observed when all these members reached the reproductive stage. Carlquist [125] correlated the formation of interxylary phloem with the flowering season of the taxon of Onagraceae. The present study also reports similar observations in *M. pruriens*. However, members of the Convolvulaceae family show a combination of multiple cambial variants. Therefore, in the present study, the development of interxylary phloem is considered only from the unlignified parenchyma patches that are enclosed in the secondary xylem. These parenchyma islands are produced by the regular vascular cambium and not by the successive cambia. During cell division and differentiation, it produces patches of thin-walled, unlignified parenchyma internally (Figure 4C). Species such as *Argyreia elliptica*, *Canavalia ensiformis*, *C. gladiata*, *Coccinia grandis*, *Hewittia malabarica*, *Ipomoea hederifolia,* and *I. turbinata* show interxylary phloem only after the initiation of the reproductive phase, although the parenchyma islands develop during the extension growth (Figure 4D). In the case of *Salvadora oleoides* and *S. persica* (Figure 4A), such interxylary phloem development can occur immediately after the formation of the parenchyma band from the cambium or later, which is a regular feature of every parenchyma island of this taxon. Over time, the associated parenchyma dedifferentiates into small segments of 2–3 cells with wide radial files of meristematic cells such as cambium (Figure 4B). New sieve elements are added by this cambium to replace the non-conducting elements. The diameters of interxylary phloem sieve elements differ from those of regular phloem and they can be oval to circular or polygonal in shape or show a combination of a square to rectangular outline in transverse view (Figure 4E). According to Carlquist [10,109], the formation of interxylary phloem facilitates the rapid translocation of photosynthates for massive flowering and the development of fruits, which is a high-energy-consuming process. Similar observations were also documented in our earlier studies [60,62,64,74]. However, our conclusion is based on field observations and no experimental evidence is available on the correlation between the reproductive phase and the formation of interxylary phloem. Besides the *Combretum* subtype (i.e., inward deposition of phloem), *Leptadenia reticulata* also shares the *Azima* subtypes' interxylary phloem development through the dedifferentiation of previously formed parenchyma islands. In the thick stems of *L. reticulata*, axial parenchyma adjacent to the xylem rays undergo repeated divisions and become meristematic and arrange the cells in radial files like cambium (Figure 4F).

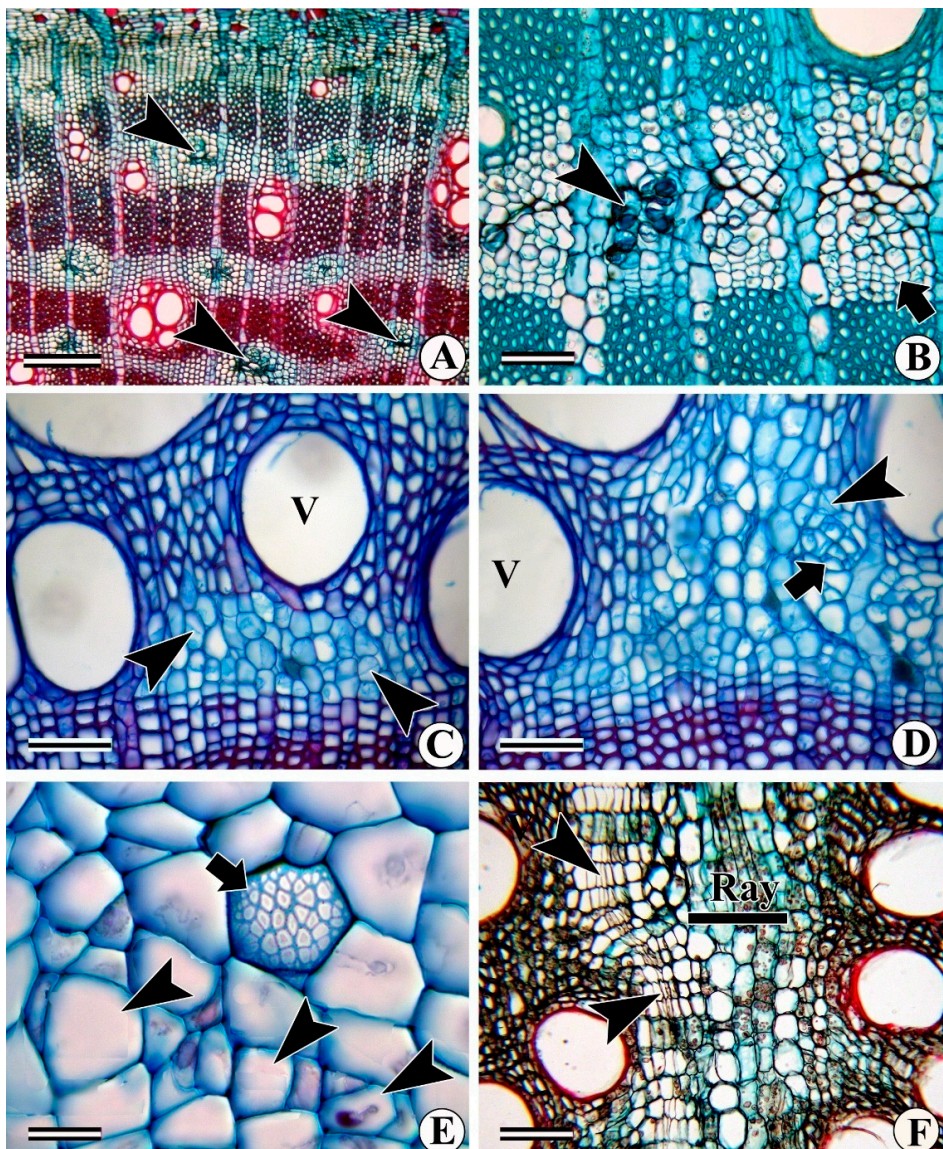

**Figure 4.** Transverse view of mature stems showing interxylary phloem. (**A**) Interxylary phloem islands (arrowheads) embedded within the secondary xylem of *S. oleoides*. (**B**) Associated parenchyma showing division and radially flattened cells like cambium (arrow). Arrowhead indicates previously formed sieve elements in *S. persica*. (**C**) Parenchyma strand in a relatively young stem showing initiation of cell division (arrowheads) and formation of sieve elements in *A. elliptica*. (**D**) Fully grown thick stem of *A. elliptica* showing recently formed sieve elements (arrow) and continuation of cell division (arrowhead) in adjacent parenchyma cell. (**E**) Structure of sieve elements in transverse view showing simple sieve plate (arrow). Note the variation in the outline shape and size (arrowheads). (**F**) Initiation of meristematic activity (arrowheads) in the parenchyma cells adjacent to the xylem rays. A horizontal bar indicates the width of the xylem ray in *L. reticulata*. *Abbreviation*: V = vessel. *Scale bars*: (**A**) = 200 μm; (**B**–**D**,**F**) = 100 μm; (**E**) = 20 μm.

In contrast, the development of interxylary phloem in the *Coccinia grandis* differs from the above types. In this case, unlignified parenchyma patches within the secondary xylem show no dedifferentiation of axial parenchyma in stems with thicknesses of up to 15 mm (Figure 5A). Stems with thicknesses of 20–25 mm or more show dedifferentiation of axial parenchyma and become meristematic (Figure 5C–F) and form several small segments of cambium with irregular orientations (Figure 5D–F).

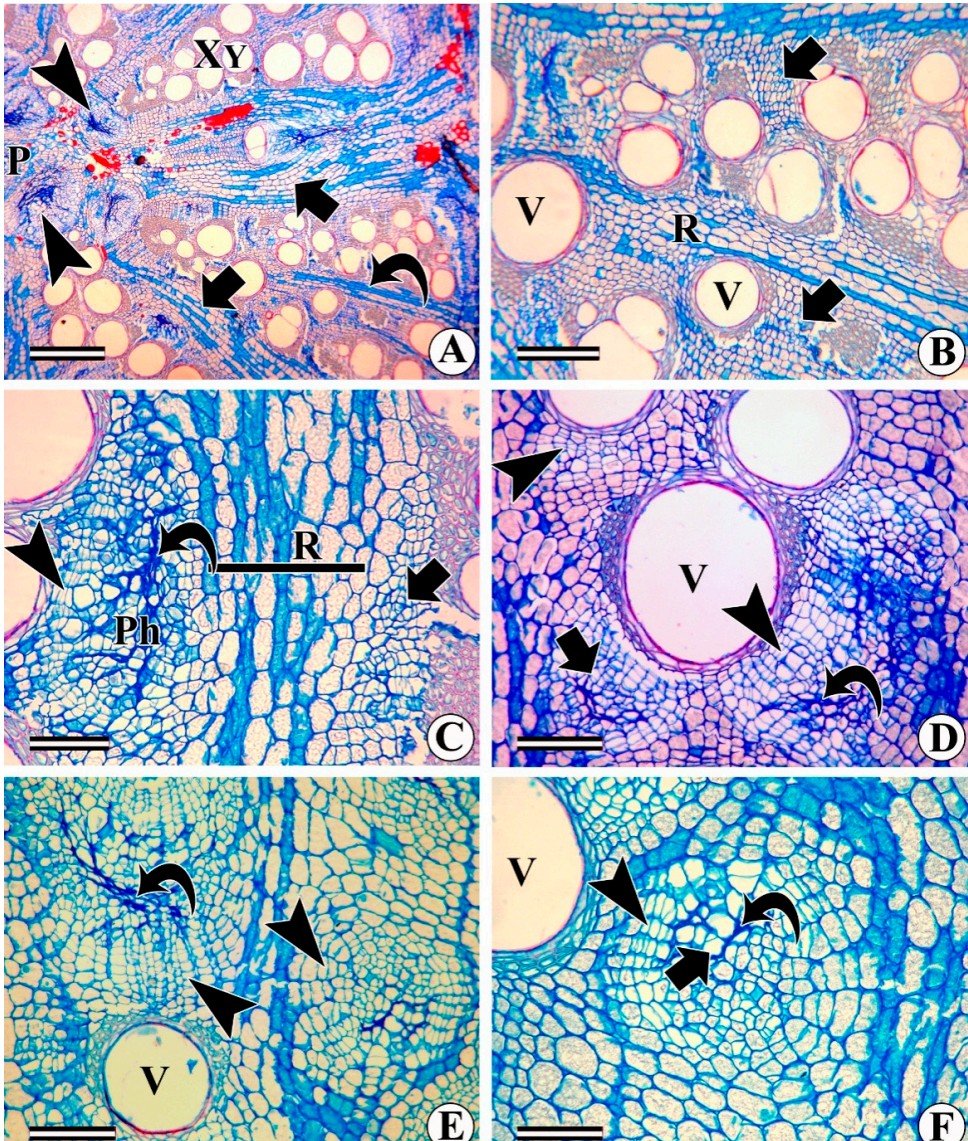

**Figure 5.** Transverse view of the thick stems of *Coccinia grandis* showing various stages of parenchyma dedifferentiation. (**A**) Structure of secondary xylem before the initiation of interxylary phloem in a 15 mm thick stem. Arrowheads indicate phloem (bicollateral) on the pith margin, arrows indicate medullary rays, and the curved arrow shows the secondary ray. (**B**) Enlarged view of Figure 5A showing the structure of the secondary xylem. Arrows indicate unlignified parenchyma patches. (**C**) Dedifferentiation of axial parenchyma adjacent to the rays forming files of meristematic cells (arrowhead). The curved arrow indicates non-conducting and crushed phloem. The arrow shows the divisions in the axial parenchyma followed by the differentiation of phloem elements. The horizontal bar (R) indicates the xylem ray. (**D**) Axial parenchyma adjacent to the vessels showing meristematic activity (arrowheads), phloem elements (arrow), and crushed phloem (curved arrow). (**E**) Part of the secondary xylem showing the meristematic activity (arrowheads) of the axial parenchyma and the formation of phloem elements. The curved arrow indicates the crushed sieve elements. (**F**) Enlarged view of Figure 5E showing the circular arrangement of meristematic cells (arrowhead), sieve element (arrow), and crushed phloem (curved arrow). *Abbreviations*: Ph = secondary phloem, P = pith, R = xylem ray, V = vessel, XY = secondary xylem. ***Scale bars***: (**A**) = 500 μm; (**B**,**E**) = 200 μm; (**C**,**D**,**F**) = 100 μm.

The formation of such meristematic segments from the axial parenchyma adjacent to the vessels and xylem rays was a common feature in the stems of all the samples investi-

gated. All these cambial segments produced only sieve elements and xylem differentiation was not observed (Figure 5C–F). Over time, previously formed sieve elements became non-conducting and underwent crushing (Figure 5C). It is interesting to note that non-conducting and collapsed sieve elements showed the presence of open sieve pores without any callose or slime deposition.

### 3.1.4. *Calycopteris* Subtype

A unique pattern of interxylary phloem development was observed in *Calycopteris floribunda* (Combretaceae). In this species, some of the small sectors of this cambium divide rapidly and form several cell-wide cambial zones (Figure 6A). Subsequently, the central cells of the cambial zone differentiate into lignified xylem derivatives, which consequently separate the wide cambial zone into outer and inner segments (Figure 6B). Afterwards, the inner cambial zone is less active/ceases to produce xylem internally and deposits only sieve elements, whereas the outer cambial segment remains in co-ordination with the adjacent cambial segments, thus enclosing the inner segment of cambium in the secondary xylem (Figure 6C). Such behaviour is repeated randomly at several sites of the cambial ring. This consequently results in a foraminate type of interxylary phloem formation. The genus *Calycopteris floribunda* belongs to Combretaceae but does not share the features of the *Combretum* subtype's interxylary phloem development [55]. Although this type of phloem development was reported in 1866 by Fritz Müller [117], we are still far from having a complete list of species with interxylary phloem based on ontogenetical events. There may be several such cases existing in nature but studies on phloem (both, regular and variant) development are rare compared to wood, which may be due to the relatively lower economic significance of phloem.

A similar but slightly different method of interxylary phloem development was observed in *Dalechampia coriacea* (Euphorbiaceae). In this case, a single ring of vascular cambium remains functional its whole life and produces secondary xylem and phloem like most eudicots. The vascular cambium divides bidirectionally to produce secondary phloem centrifugally and lignified thick-walled xylem derivatives centripetally. During this activity, the cambium also deposits a tangential band of unlignified, thin-walled axial parenchyma cells that is a few cells wide. Thus, structurally, the secondary xylem shows the presence of tangential bands of thin-walled, unlignified parenchyma that initially lack interxylary phloem in the young stems (Figure 7A,B). With the increase in stem thickness, some of these parenchyma cells dedifferentiate and show repeated periclinal divisions, forming a layer of the meristematic zone that is a few cells wide (Figure 7B). Subsequently, their number increases and becomes a cambium-like zone several cells wide that possesses radial files of meristematic cells (Figure 7C). Concomitantly, some of the cells positioned centrally in the cambial zone begin to differentiate into sieve elements that have been intermixed with the sclerenchymatous cells (Figure 7C) by leaving tangential bands of meristematic cells on either side (Figure 7C–E). Some of the cambial cells also differentiate into xylem derivatives (Figure 7D,F). It is dubious to keep this part in the interxylary phloem section. Although it gives an appearance like successive cambia (Figure 7D,E), they are small segments within the parenchyma strands and they do not form a complete ring (Figure 7A). A unique case of interxylary phloem development in *Dalechampia coriacea* (Euphorbiaceae) does not fit into any of the categories mentioned above and has not been reported so far in the literature. In this species, radial files of tangential bands, a few cells wide, of the parenchyma cells differentiate internally within the secondary xylem. Subsequently, the marginal cells of these parenchyma bands become meristematic and deposit sieve elements and the development of the xylem is observed occasionally.

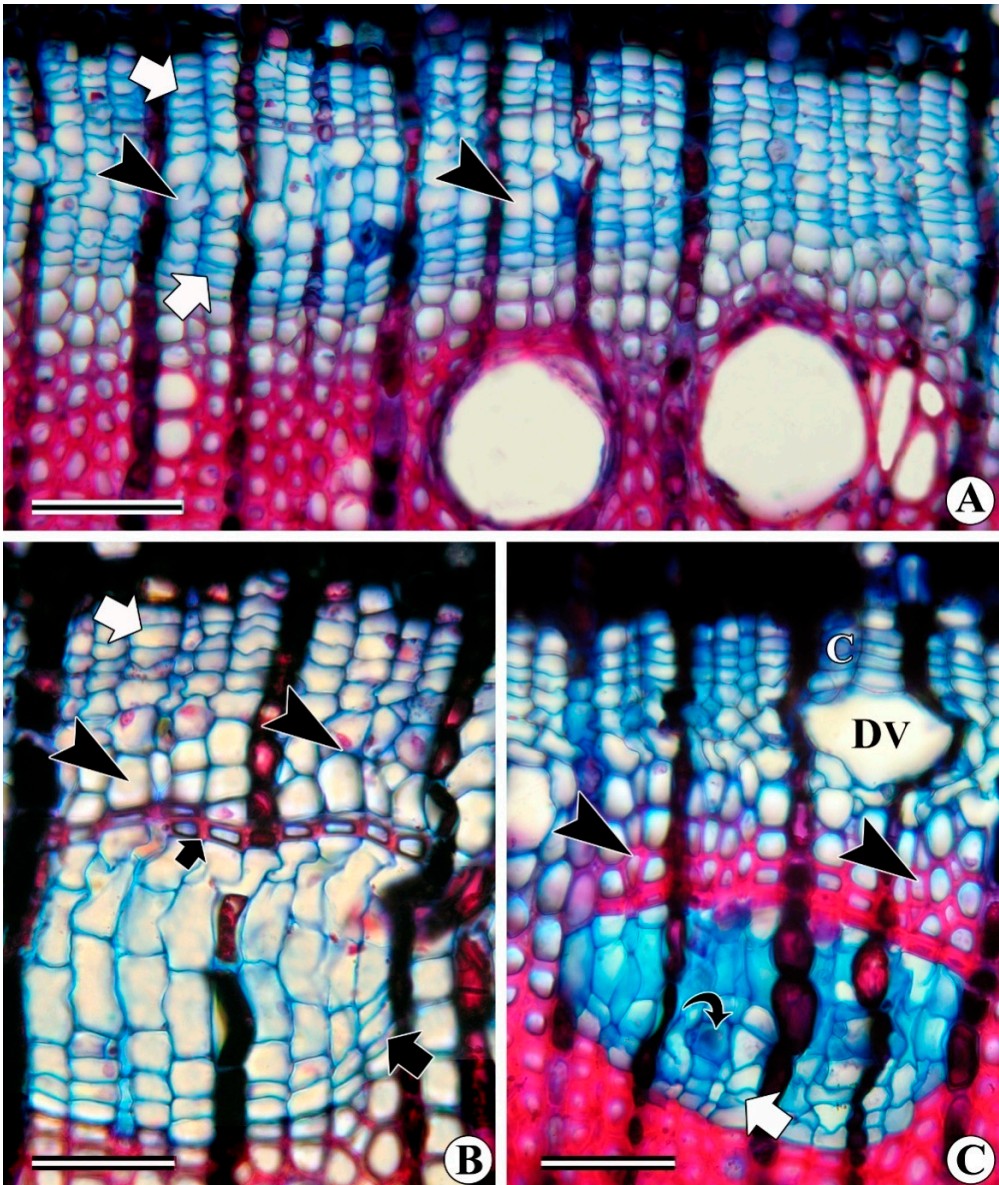

**Figure 6.** Transverse view of *Calycopteris floribunda* stems showing the development of interxylary phloem. (**A**) Initiation of separation of the cambium into outer and inner strips (arrows). Note the cambial zone from the left corner of the image to the right showing all the stages of cambium separation. Arrowheads indicate differentiating xylem derivatives. (**B**) Widely separated outer and inner cambial segments (arrows) and differentiating thin-walled xylem derivatives (arrowheads). Note that cambial cells transformed into thick-walled xylem derivatives (small arrow). (**C**) Initiation of deposition of lignified elements (arrowhead) and inclusion of the separated cambium along with the phloem within the secondary xylem. The curved arrow indicates the differentiating sieve element. Arrow indicates radially arranged meristematic (cambium) cells. *Abbreviations*: C = cambium, DV = differentiating vessel. ***Scale bars***: (**A**) = 100 μm; (**B**,**C**) = 50 μm.

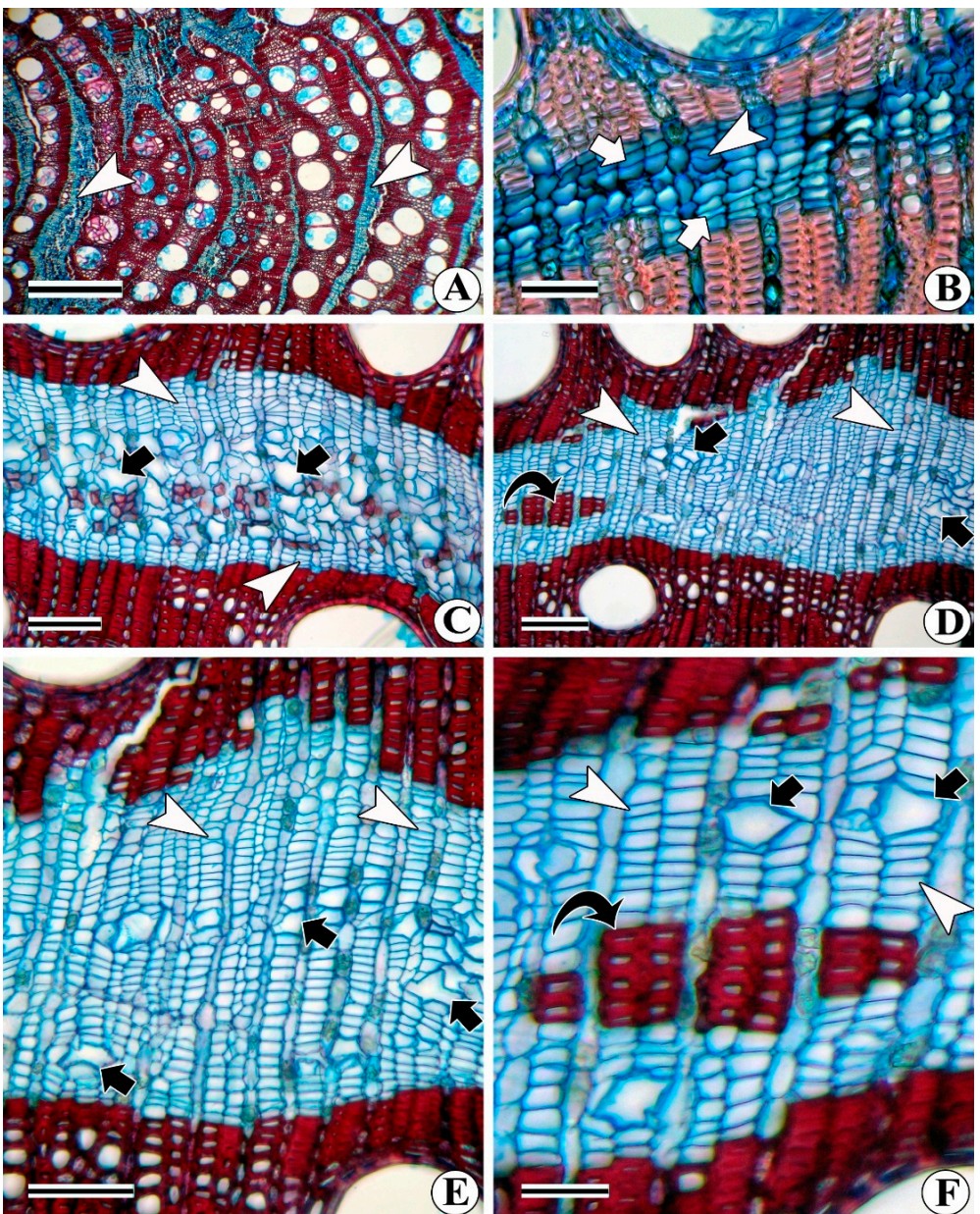

**Figure 7.** Transverse view of *Dalechampia* stems showing various stages of interxylary phloem development. (**A**) A mature stem showing tangential bands of unlignified axial parenchyma (arrowheads). Note the radial width of the parenchyma band (left arrowhead). (**B**) Enlarged view of one of the parenchyma bands showing the arrangement of cells (arrows). Note the dividing cell (arrowhead). (**C**) Differentiation of sieve elements (arrows) and separation of cambium (arrowheads) due to the formation of phloem cells. Note the safranin-stained sclerenchymatous cells adjacent to the phloem. (**D**) Several cells wide, radially arranged cambial cells (arrowheads). Note the differentiated phloem (arrows). The curved arrow shows lignified xylem elements. (**E,F**) Enlarged view of Figure 7D showing radial files of cambial cells (arrowheads), phloem elements (arrows), and xylem elements (curved arrow) in Figure 7F. *Scale bars*: (**A**) = 500 μm; (**B,F**) = 50 μm; (**C–E**) = 100 μm.

Interxylary phloem islands do not form continuous strands from the base of the stem up to the tip; rather, they anastomose to form a network within the secondary xylem. In the case of widely separated phloem islands, the development of radial sieve elements is a common feature that interconnects them to form a network in all the subtypes of the interxylary phloem. Details of the radial sieve elements' formation are discussed later in a separate section. Regarding the function of the interxylary phloem, there is ambiguity and no unique

functions have been ascribed to it in the existing literature. According to Carlquist [109], studies on the descriptive anatomy of interxylary phloem and allied phenomena, such as intraxylary phloem and bicollateral vascular bundles are incomplete. Therefore, the interpretation of interxylary phloem's functions is intricate since its occurrence is reported in a small number of families [109]. It performs the function of photosynthate translocation since it is protected within the fibrous tissue. It is embedded within the secondary xylem and may be associated with protection from insect damage or protection of the phloem tissue from harsh environmental conditions. However, experimental studies are needed to confirm this hypothesis.

### 3.2. Intraxylary (Internal/Medullary) Phloem

The development of intraxylary phloem is another important aspect of plant growth and development and is an alternate pathway for the transport of photosynthates and several other compounds. This development occurs on the inner margin of the protoxylem elements, which are categorized into three types and come from (i) the procambial derivatives (e.g., *Ipomoea hederifolia*, *I. muricata*, *Turbina corymbosa*). Strands of phloem that develop on the pith margin, i.e., on the inner side of the protoxylem, are referred to as intraxylary phloem. Their occurrence has been reported in 19 families of eudicots [81]. Ontogenetically, their development differs from species to species and occurs before, simultaneously, or after the formation of regular protophloem and may be produced from the ground meristem ([30] p. 362) pericycle, or procambial derivatives [8,126–128] or from the marginal pith cells [10,109,129].

Intraxylary protophloem can develop simultaneously, before, or after the formation of protoxylem elements (Figure 8A–F) (e.g., *Ipomoea* sps., *Turbina corymbosa*, *Solanum pseudocapsicum*, *Vallaris solanacea*); (ii) the mature pith cells (*Argyreia nervosa*, *Argyreia* sp., *Baeumontia jerdoniana*), which may be adjacent to the protoxylem or away from it (Figure 9A–E); and (iii) the intraxylary phloem cambium also called the medullary cambium (e.g., *I. hederifolia*, *I. muricata*, *T. corymbosa*, *Campsis radicans*, *Distimake tuberosus*, *Solanum pseudocapsicum*, *Hewetia malabarica*, etc.).

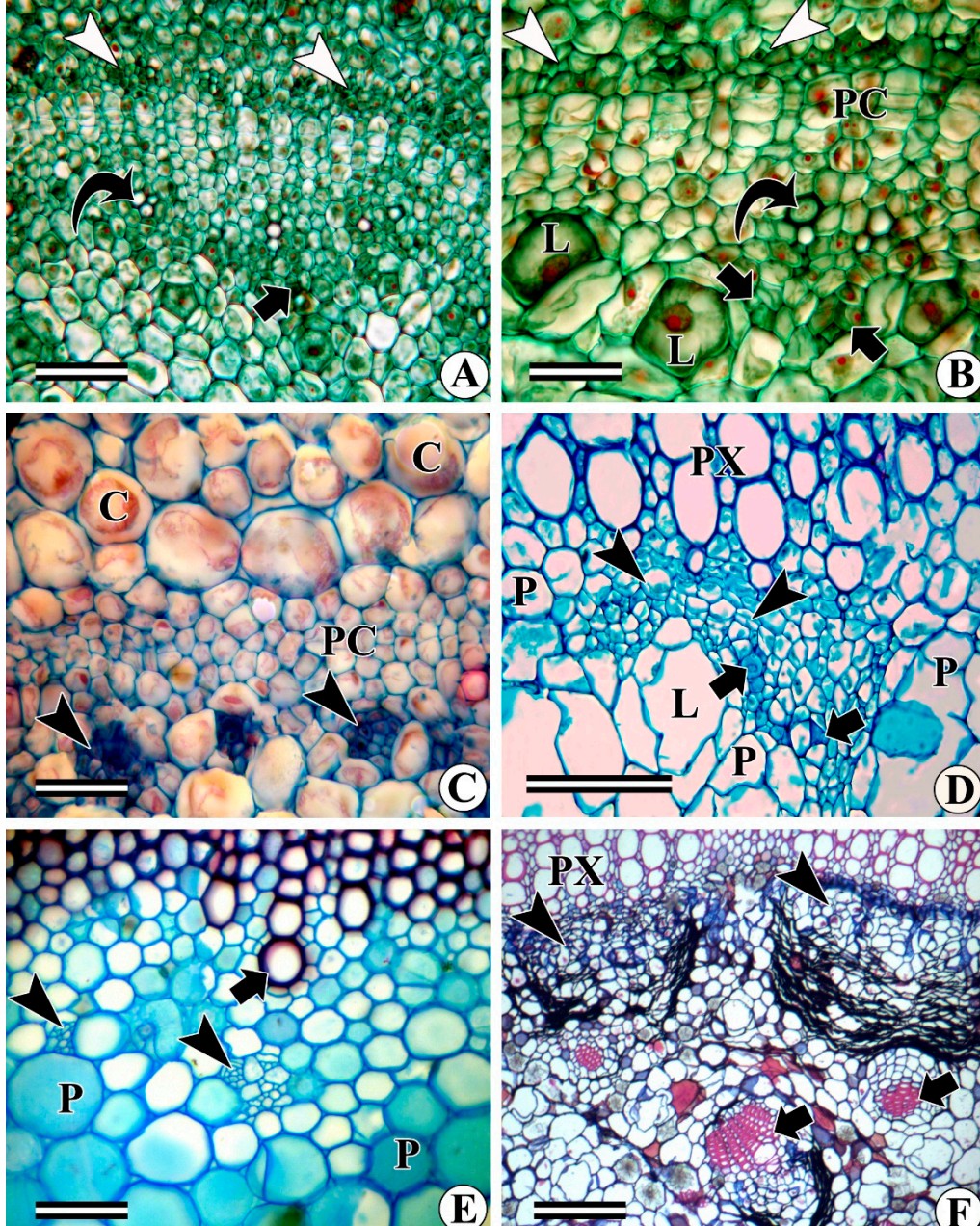

**Figure 8.** Development of intraxylary protophloem in young stems. (**A**) Initiation of intraxylary protophloem development (arrow) in the young stem of *Ipomoea hederifolia*. Arrowheads indicate external protophloem and the curved arrow shows the protoxylem. (**B**) Enlarged view showing differentiating intraxylary protophloem (arrows). Arrowheads indicate external protophloem and the curved arrow shows the protoxylem. (**C**) Development of intraxylary phloem (arrowheads) precedes that of regular protophloem in *Solanum pseudocapsicum*. Note the absence of regular (external) protophloem. (**D**) Development of additional intraxylary phloem (arrows) from the procambial cells in *I. hederifolia*. Arrowheads indicate dividing procambial derivatives. Note the variation in the size of the cells adjacent to the protoxylem (PX) elements. (**E**) Young stem of *Argyreia nervosa* showing intraxylary phloem (arrowheads) in the pith region and away from the protoxylem (arrow). (**F**) Fully grown thick stem of *A. nervosa* showing intraxylary phloem (arrowheads) on the inner margin of the protoxylem (PX). Arrows indicate medullary bundles. *Abbreviations*: C = cortex, L = laticifers, P = pith cells, PC = procambium, PX = protoxylem. ***Scale bars***: (**A,E**) = 100 μm; (**B**) = 50 μm; (**C,D**) = 75 μm; (**F**) = 200 μm.

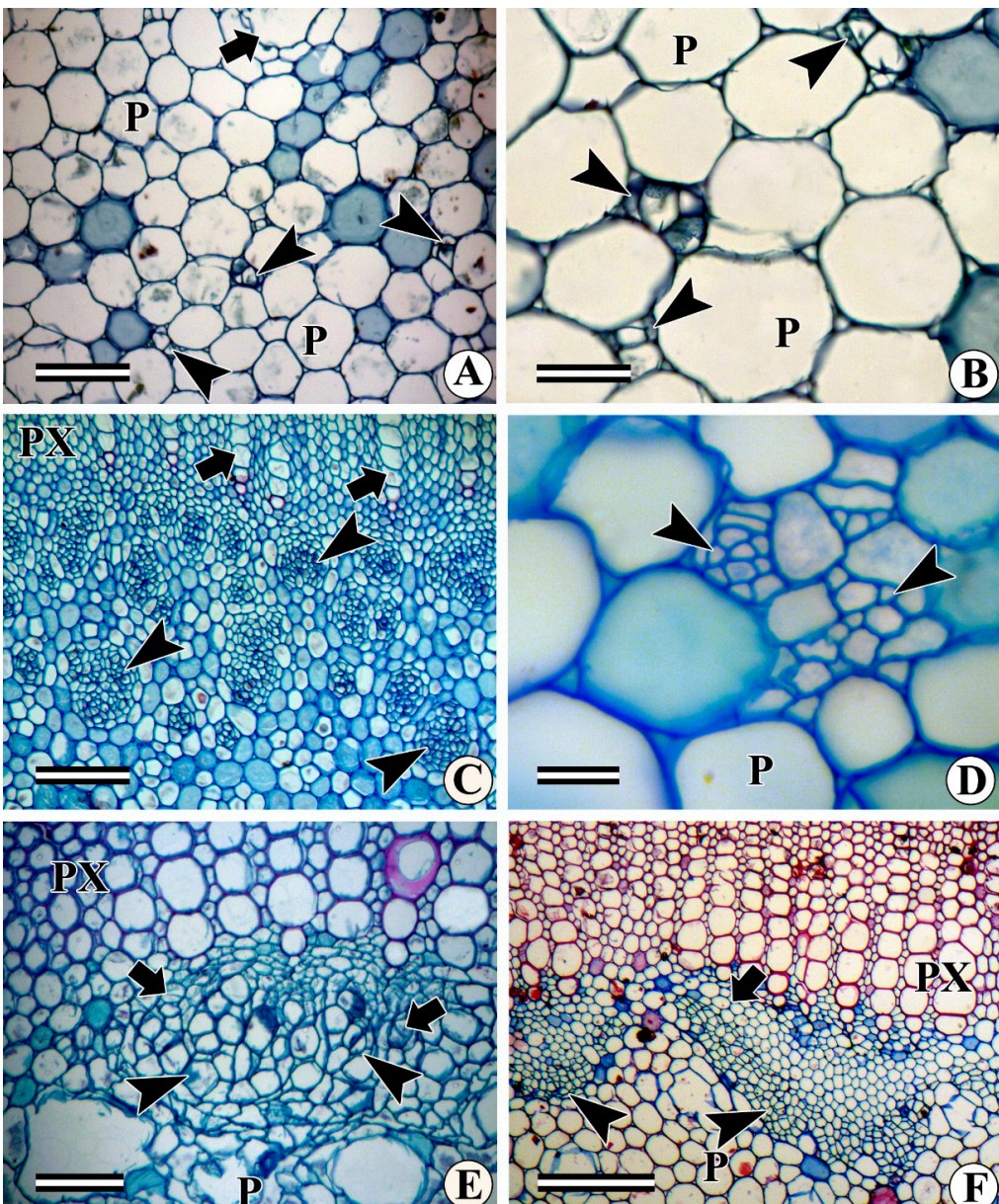

**Figure 9.** Transverse view of the pith portion of the stems showing intraxylary phloem distribution. (**A**,**B**) Isolated strands of intraxylary phloem (arrowheads) in the pith portion (**A**) and enlarged view (**B**) of *Argyreia sericea*. Arrow indicates laticifers. (**C**) Distribution of intraxylary phloem (arrowheads) in the pith of *Beaumontia jerdomiana*. Arrows show protoxylem. (**D**) Enlarged view of Figure 8E showing the morphology of sieve elements (arrowheads). (**E**,**F**) Intraxylary (secondary) phloem showing clusters of the sieve elements (arrowheads). Note the dividing peri-medullary cells (arrows) in *Merremia hederacea* (**E**) and *A. osyrensis* (**F**). *Abbreviations*: P = pith cells, PX = protoxylem. **Scale bars**: (**A**,**C**,**E**) = 100 m; (**B**) = 50 μm, (**D**) = 20 μm, (**F**) = 200 μm.

In *Beaumontia jerdomiana* in the species of *Ipomoea*, *Turbina corymbosa*, etc., the development of intraxylary protophloem takes place concomitantly with the differentiation of external (regular) protophloem and protoxylem (Figure 8A,B), whereas in S. pseudocapsicum, it occurs before the deposition of regular external phloem (Figure 8C). In most of the members of the Convolvulaceae family, the development of the intraxylary phloem occurs concomitantly with the protoxylem but it deviates in *A. nervosa*. This species shows the absence of an intraxylary protophloem but the development of secondary intraxylary phloem from the mature pith cells and intraxylary phloem cambium. With the initiation of

the secondary growth stage, several species show the deposition of additional secondary intraxylary phloem from the adjacent parenchymatous pith cells (Figure 8D). In the transverse view, these cells differ in shape and size from the abutting pith cells and their shape and dimensional details vary from species to species. However, different species of *Argyreia* show variations in the development of intraxylary phloem. In *A. sericea* (Figure 9A,B), although it develops concurrently with the regular protoxylem and protophloem, it is significantly further away from the pith margin (Figure 8E). In contrast, *A. nervosa* shows the formation of the medullary bundles instead of the intraxylary phloem (Figure 8F).

However, its development in *A. nervosa* begins close to the protoxylem only in the relatively thick stems after the initiation of the secondary growth stage (described later along with medullary bundles). *A. sericea* shows isolated strands of 2–3 sieve elements distributed randomly throughout the pith region (Figure 9A,B). On the other hand, *Beaumontia jerdoniana* shares both kinds of locations of intraxylary phloem deposition, i.e., on the inner margin (adjacent) of the protoxylem and significantly further away from it (Figure 9C). In this case, it forms islands of several sieve elements in groups that are distributed randomly throughout the pith region.

The sieve elements of the intraxylary protophloem can be several times smaller than the adjacent pith cells and regular external phloem. They are oval-elliptic or rectangular, containing one to many sieve elements within the small area that is equivalent to the adjacent single pith cells (Figures 8E and 9D). As the secondary growth stage progresses, additional sieve elements differentiate from the mature pith cells (Figure 9E). The development of sieve elements was observed in several species (e.g., *A. osyrensis*, *Beaumontia jerdomiana*, *Camonea vitaefolia*, *Hewetia malabarica*, *I. hederifolia*, *I. muricata*, *Merremia hederacea*, *Solanum pseudocapsicum*, *T. corymbosa*, etc.). Depending on the species, structurally marginal pith cells that give rise to the secondary intraxylary phloem vary in shape and size (Figure 9E,F). These marginal pith cells are dedifferentiated into a cluster of sieve elements (Figure 9E,F).

**Development of intraxylary phloem cambium (medullary cambium)***: In relatively thick stems (as per our phenological observations when the plant enters the reproductive phase), the marginal pith cells become meristematic and divide periclinally to form radial files of cells (Figure 10A–D) that share all the characteristics of the vascular cambium. This is variously referred to as internal cambium, intraxylary phloem cambium, and medullary cambium/peri-medullary cambium. the cell division activity of the intraxylary phloem cambium differs from species to species and sometimes it becomes species-specific. In most species, it deposits only phloem derivatives (e.g., *Argyreia elliptica*, *A. nervosa*, *Dicranostylis ampla*, *Solanum pseudocapsicum*) even in fully grown thick stems (Figure 10C,D). In contrast, in species such as *Campsis radicans*, *Hewettia malabarica*, *I. hederifolia*, *I. muricata*, *Turbina corymbosa*, *Vallaris solanacea*, and several other members of the Convolvulaceae family, this cambium is functionally bidirectional and produces both secondary xylem centrifugally and secondary phloem centripetally (Figure 10C–F, 11A–C). Depending on the species, the ratio of the xylem to phloem production from the intraxylary phloem cambium differs from species to species. Some species such as *Campsis radicans* (Figure 11A,B) and *Hewettia malabarica* (Figure 11C) show the deposition of more xylem than phloem. Products of this cambium consist of wide and narrow (fibriform) vessels, fibres, and rays in the xylem (Figure 11B), whereas the phloem is composed of sieve elements, companion cells, and associated axial and ray parenchyma cells (Figure 10C). Some samples of Ipomoea turbinata show a unique behaviour of the intraxylary phloem cambium, where it is functionally unidirectional and produces both secondary xylem and phloem centripetally, i.e., in the same direction (Figure 11D,E). Some of the samples of the same species (i.e., *I. turbinata*) also show the presence of two intraxylary cambia on either side of the lignified tissue, in which the outer (adjacent to protoxylem) cambium deposits secondary phloem externally and xylem internally, whereas the inner cambium produces phloem centripetally and xylem centrifugally (Figure 12A). Structurally, the phloem remains similar to the regular external

phloem and can possess simple or compound sieve plates that are oriented transversely or obliquely to the main axis (Figure 12B).

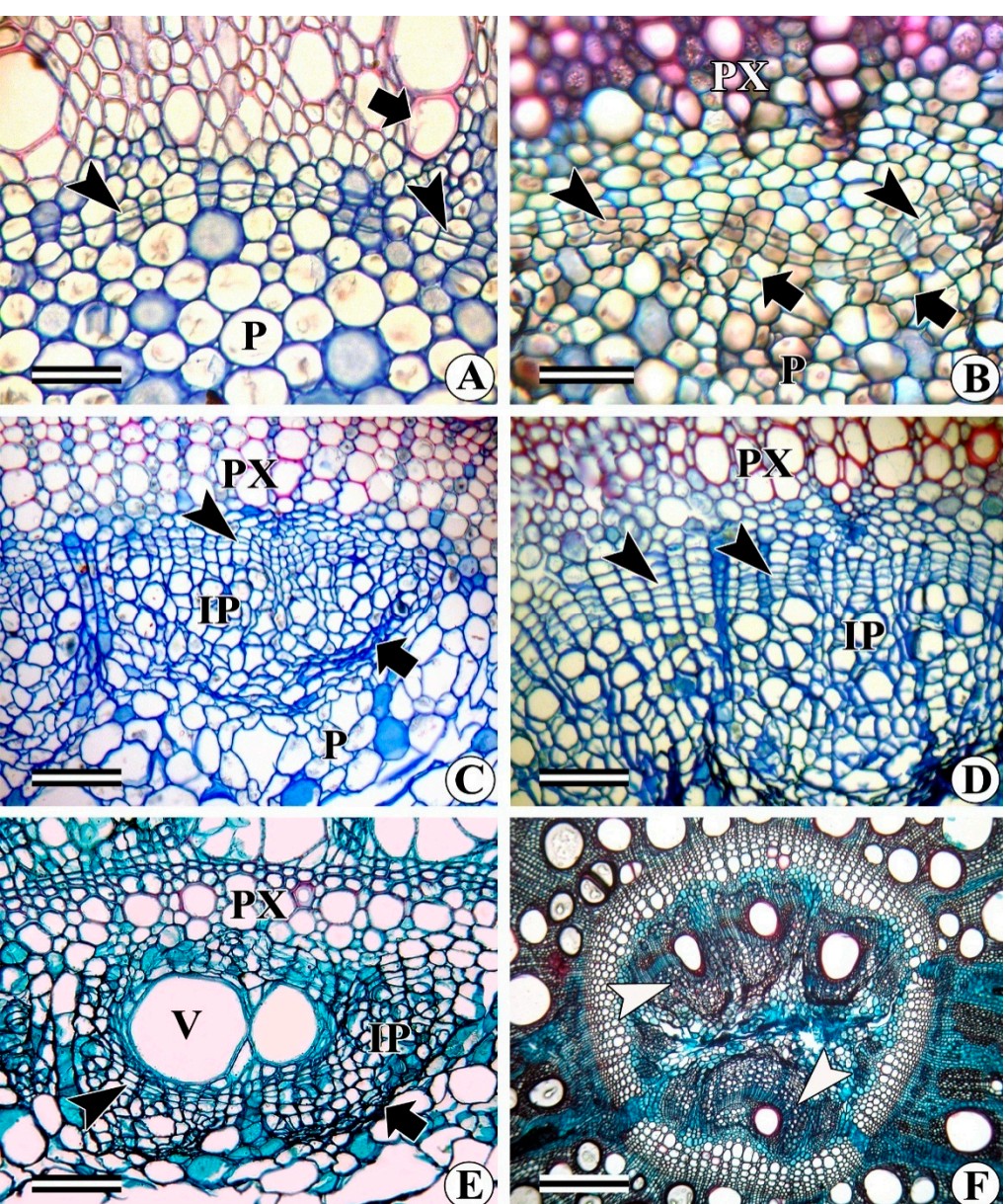

**Figure 10.** Transverse view of the stems of different species showing the initiation of intraxylary phloem (peri-medullary) cambium. (**A**) Initiation of intraxylary phloem cambium development in *Beaumontia jerdomiana*. Note the peri-medullary cells showing the initiation of cell division (arrowheads). The arrow shows the protoxylem vessel. (**B**) Intraxylary phloem cambium (arrowheads) and its derivatives in *Campsis radicans*. Arrows indicate sieve elements. (**C**) Unidirectional activity of intraxylary phloem cambium in *Dicranostylis ampla* showing only phloem (IP) formation. Arrowhead indicates cambium and the arrow shows crushed intraxylary phloem. (**D**) Intraxylary phloem cambium (arrowheads) showing cells arranged in radial files and phloem (IP) deposited by the cambium in *Merremia dissecta*. (**E**) Bidirectional activity of intraxylary phloem cambium in *Ipomoea hederifolia*. Arrowhead indicates intraxylary phloem cambium and the arrow indicates crushed intraxylary phloem. (**F**) Bidirectional activity of intraxylary phloem cambium showing secondary xylem and phloem (arrowheads) deposited in the pith. *Abbreviations*: P = pith, PX = protoxylem, IP = intraxylary phloem, V = vessel. ***Scale bars***: (**A**) = 50 μm, (**B**–**E**) = 100 μm, (**F**) = 500 μm.

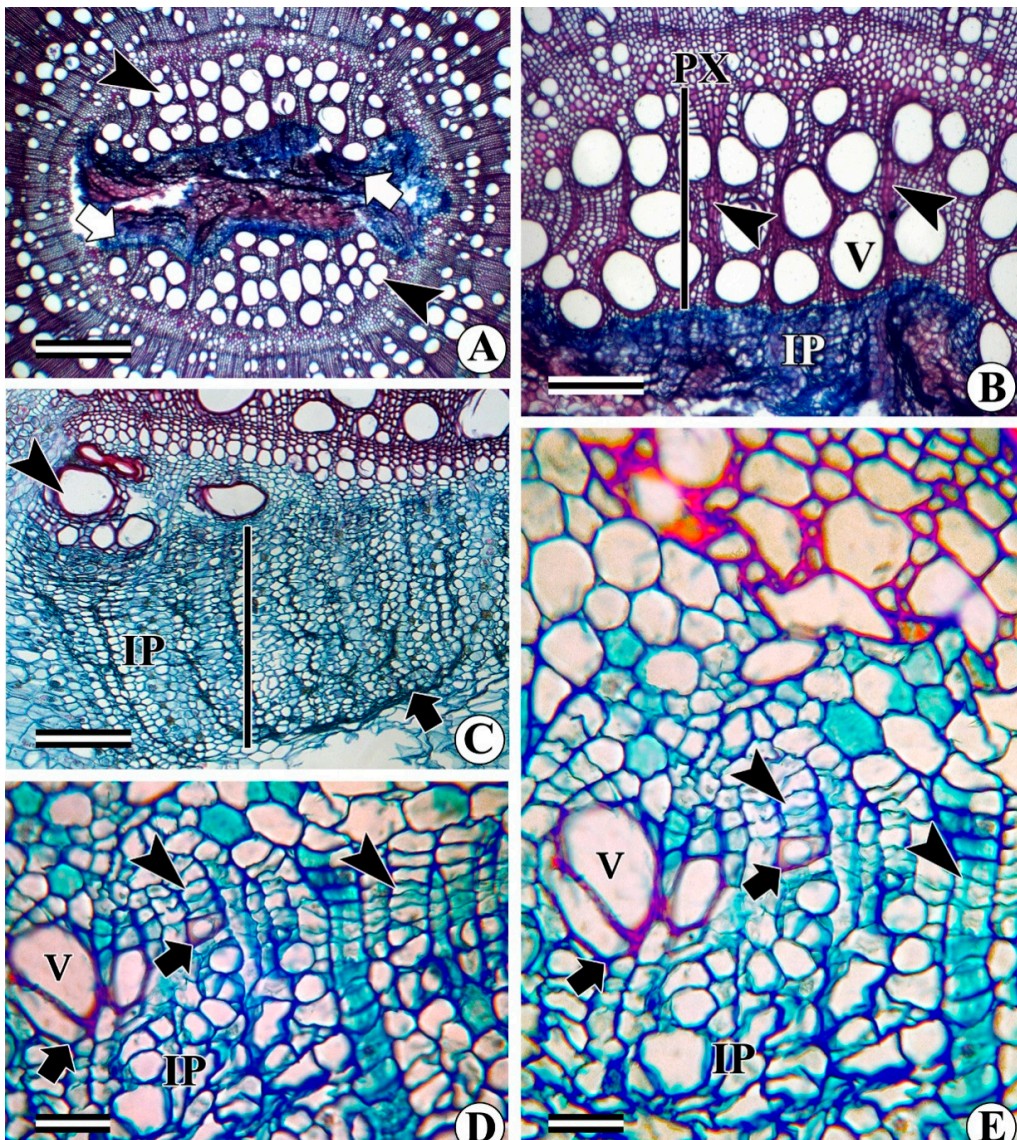

**Figure 11.** Intraxylary phloem cambium and its products in different species of eudicots. (**A**) Bidirectional activity of the intraxylary phloem cambium that deposits a measurable quantity of secondary xylem (arrowheads) and phloem (arrows). Note that the pith is completely occupied by its products in. (**B**) Enlarged view of Figure 11A showing the structure of the xylem formed by intraxylary phloem cambium. The vertical line indicates the quantity of xylem formed by the intraxylary phloem cambium. Arrowheads indicate the ray-like arrangement of parenchyma cells. (**C**) Development of more phloem (IP) than secondary xylem (arrowhead) in Hewittia malabarica by intraxylary phloem cambium. Arrow indicates crushed non-conducting phloem. Note the amount of phloem (vertical bar). (**D**) Unidirectional differentiation of both xylem and phloem by the intraxylary phloem cambium. Note the position of intraxylary phloem cambium (arrowheads), xylem (arrows), and phloem (IP) in *Ipomoea turbinata*. (**E**) Enlarged view of Figure 11D. Abbreviations: IP = intraxylary phloem, PX = protoxylem, V = vessel. ***Scale bars***: (**A**,**D**,**E**) = 500 μm, (**B**,**C**) = 200 μm.

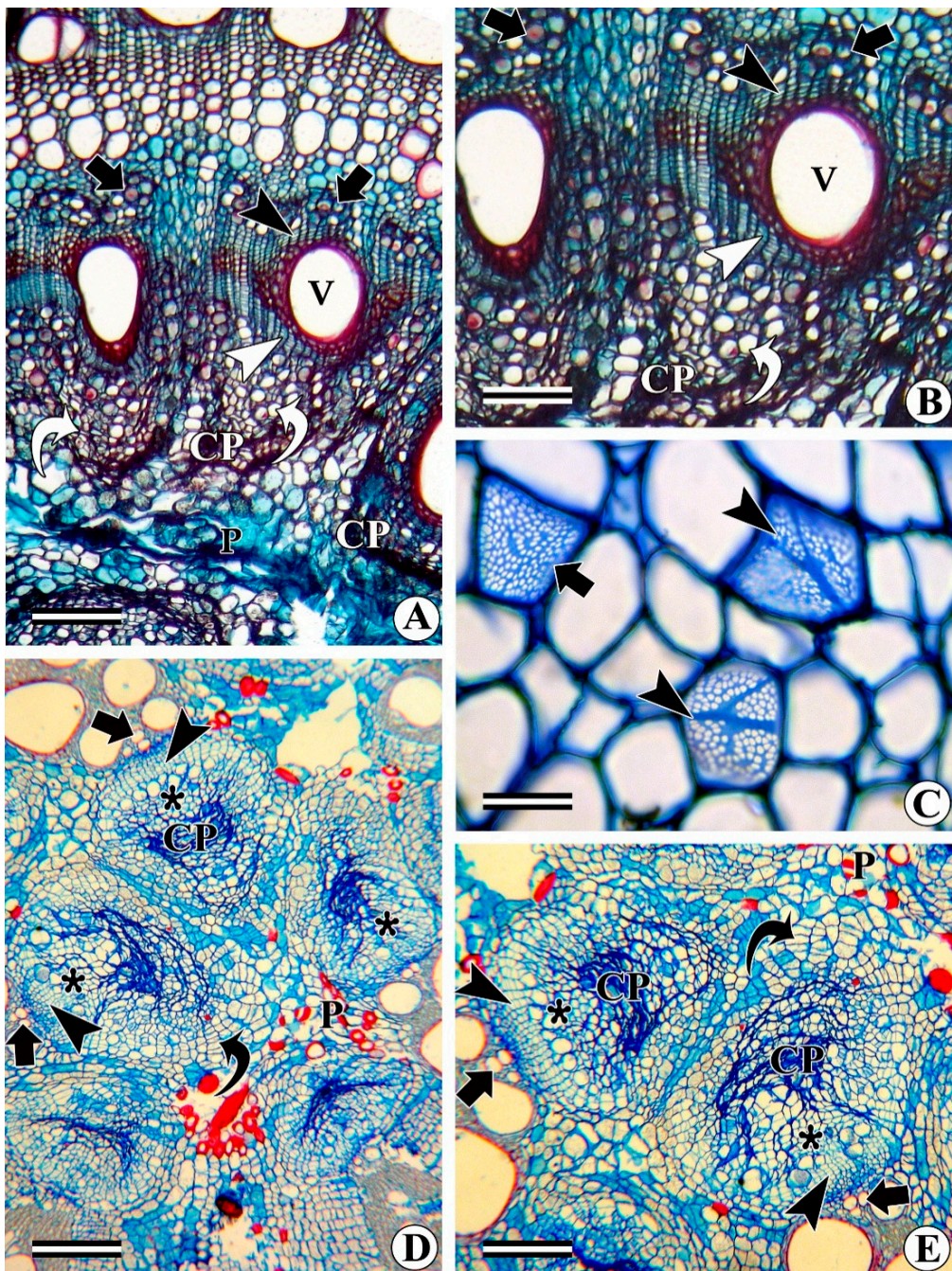

**Figure 12.** Transverse view of mature stems showing intraxylary phloem cambium (**A**,**B**), the structure of sieve elements, and phloem of bicollateral vascular bundles. (**A**) Intraxylary cambia in *Ipomoea turbinata.* An outer cambium (black arrowhead) and phloem developed from it (black arrow). An inner cambium (white arrowhead) and phloem formed from it (curved arrow). (**B**) Enlarged view of *I. turbinata.* Figure legend follows a similar indication. (**C**) Structure of sieve plates in *I. turbinata.* Note the differences (arrows and arrowheads). (**D**) Bicollateral vascular bundles in thick stems of *Coccinia grandis* showing additional secondary phloem (asterisk) developed on the margin of the protoxylem (arrow). Intraxylary phloem cambium (arrowheads) and meristematic rings that formed later that encircle the phloem (curved arrows). (**E**) Enlarged view of Figure 12D. Figure legend follows a similar indication. *Abbreviations*: CP = crushed intraxylary phloem; P = pith cells, V = vessel. **Scale bars**: (**A**,**D**) = 200 μm, (**B**,**E**) = 100 μm, (**C**) = 20 μm.

As the secondary growth stage progresses, the bulk of the additional secondary intraxylary phloem deposition can take place from the adjacent pith cells or by the cambial

action at the pith margin, as reported in some species [10,60,74,124,130]. The initiation of the intraxylary phloem cambium on the pith margin is a common feature in the families like Convolvulaceae, Solanaceae, and Apocynaceae [8,49,60,62,74,94,124,130]. The initiation of the cambium at the pith margin has been known for more than a century and is variously referred to as internal cambium, intraxylary phloem cambium or medullary cambium. In most species, the intraxylary phloem develops from the functionally unidirectional cambium and deposits only phloem elements and no xylem. Since it produces only phloem tissue in the centripetal direction, it was variously referred to in the earlier literature as *false cambium* [94], *local cambium* [4], *inner cambium* [131], and unilateral cambium [132]. Functionally, it was found to produce only phloem and no xylem and the term 'unilateral cambium' appears to be suitable due to the development of phloem only in one direction. However, recently there have been several reports on the bidirectional activity of the intraxylary phloem cambium in different members of eudicot including most members of the family Convolvulaceae [10,49,56,60,62,74,124,129,130], as well as Apocynaceae [56], that produce xylem centrifugally and phloem centripetally. In contrast, intraxylary phloem cambium can also form phloem on two sides with a centrally located xylem, as observed in freshly collected samples of *I. turbinata* [128].

In a real sense, functionally unidirectional intraxylary phloem cambium is referred to as a condition where the secondary xylem and phloem are deposited in the same direction [128,133]. The formation of secondary xylem and phloem by the intraxylary phloem cambium in the same direction has so far only been reported in *Hoya carnosa* and *Marsdenia tomentosa* by Handa [133] and in *Ipomoea turbinata* by Rajput and Gondaliya [128]. It is a very rare phenomenon and needs further confirmation in a greater number of species. In our opinion, it may be a state of morphogenesis (i.e., autapomorphy), which is not a conserved or consistent characteristic of a particular species at a given time due to unknown reasons. This statement may be supported by the fact that due to the unidirectional differentiation observed in *I. turbinata*, additional samples of the same species were collected from different regions. A study of these additionally collected samples revealed the absence of the unidirectional nature of the intraxylary phloem cambium. In contrast, these samples showed the formation of phloem on both sides of the intraxylary phloem cambium [128]. These findings can be supported by the results obtained by Fukuda [8] who reinvestigated fully grown (mature) plants of the species studied by Handa [133]; however, Fukuda [8] failed to find the unidirectional differentiation of both kinds of vascular elements.

The gourd family (Cucurbitaceae) is known for the presence of bicollateral vascular bundles (i.e., a vascular bundle having primary phloem on the outer and inner sides of the xylem). Its occurrence is reported only during the primary growth stage but information on the deposition of additional secondary phloem on the inner margin is lacking. In the fully grown thick stems of Coccinia grandis, the associated parenchyma that encircles the inner phloem becomes meristematic and forms a complete or partial ring of cambium (Figure 12D,E) around the inner phloem of each bicollateral vascular bundles. These newly formed cambial rings deposit secondary phloem in the centripetal direction, whereas previously formed non-conducting sieve elements collapse and undergo crushing (Figure 12D,E). Since all the vascular bundles in *C. grandis* are widely separated by large medullary rays from the primary growth stage, they remain separated from each other even after secondary growth including in the 25–30 mm thick stems.

In general, several researchers use the term bicollateral vascular bundles for the intraxylary phloem located on the inner margin of the protoxylem in families such as Convolvulaceae, Solanaceae, Apocynaceae, and other families (specifically Myrtales) that show the presence of intraxylary phloem. Based on Solereder's [114] work, Scott and Brebner [5] reported the occurrence of bicollateral vascular bundles in 18 orders. The term bicollateral vascular bundle was used for the first time by De Bary in [22], although the bicollateral vascular bundle was discovered in 1854 by Hartig [18] in *Cucurbita* and it was suggested that in some plant groups, internal strands appear independent, whereas in 1855, Von Mohl [19] reported it in Asclepiadaceae and other plants (c.f., [4]). In contrast,

Hérail [118] strongly objected that true bicollateral vascular bundles are present only in the family Cucurbitaceae, whereas in other families, this phloem arises from the pith. Hérail [118] rejected the terminology of De Bary [22] on the basis that the medullary phloem does not appear at the same time as other vascular bundles and does not proceed from the same bundle (c.f., [4], p. 264). This statement can be supported by the fact that in Argyreia nervosa, intraxylary phloem development takes place only after the initiation of secondary growth [50]. The available literature also indicates that the development of primary intraxylary phloem may occur simultaneously, before, or after the development of the regular (external) and intraxylary protophloem in Apocynaceae, Convolvulaceae, Solanaceae, etc. There is enough evidence available in the literature to show that in these families, an intraxylary phloem cambium initiates between protoxylem and intraxylary phloem. In contrast, 30–35 mm-thick stems of *Coccinia grandis* [58] and *Zanonia indica* (unpublished data) fail to show the development of such intraxylary phloem cambium on the inner margin of protoxylem elements and inner phloem. It needs special mention that in the 30 mm thick stem of C. grandis [58], Figure 1D, a complete circular ring of meristematic tissue appears around the phloem that encircles the inner phloem of the vascular bundle and not a small segment of the cambium initiated, as observed in members of the Convolvulaceae family [10,49,56,60,62,74,124,129,130], *Solanum pseudocapsicum* [70] of the Solanaceae family, and *Campsis radicans* of the Bignoniaceae family [56]. This indicates that there is a difference between the intraxylary phloem of other families and that of the Cucurbitaceae family, which initiates on the inner margin of the protoxylem and bicollateral vascular bundles. The authors of the present work also agree with the opinion of Hérail [118] based on the arrangement of vascular tissues in the family Cucurbitaceae.

For instance, members of the Cucurbitaceae family show distinct vascular bundles at the beginning, followed by another set of vascular bundles that initiates externally, which alternate with the previously formed ones. All of them are distinctly separated from each other and over time, these bundles join to form a single ring of vascular cambium that is arranged in two rings. In contrast, other families that are characterized by the presence of intraxylary protophloem possess vascular strands of protoxylem and protophloem, which are not distinctly separated like those in the Cucurbitaceae family's members. Although Scott and Brebner ([5], p. 265) used the term bicollateral vascular bundles for other families, they also admitted that in the stem of the majority of eudicots, the limit between bundles is impossible to trace. Carlquist [109] also mentioned that, although "the term specifically excludes the presence of secondary xylem in such bundle, it is more commonly applied when there is little or no secondary growth in the bundle as in *Cucurbita* and *Solanum*". Unlike other families, Cucurbitaceae has distinct vascular bundles; thus, we believe that the term intraxylary phloem can be used for families other than Cucurbitaceae that show a phloem on the periphery of the pith. Like interxylary phloem, intraxylary phloem is protected (possibly from herbivory and external factors) by enclosing the phloem tissue within the secondary xylem and forming a continuous column from source to sink [60,62,124].

In addition to inter- and intraxylary phloem, the occurrence of extra-fascicular phloem has been reported in the family Cucurbitaceae by Fischer [134]. Although extra-fascicular phloem does not have direct relevance in the present study, functional differences based on its position have been reported in *Cucurbita* by Turgeon and Oparka [47]. Experimental evidence (based on gel analysis and nano-LC/MS/MS) of two different phloem types (i.e., fascicular and extra-fascicular) showed that extra-fascicular phloem plays a vital role in the conduction of several other molecules, whereas the sugar content was nearly 30% less than the required rate of photosynthate transport [40,47]. Biomolecules that are conducted within the intraxylary phloem include amino acids, proteins, macromolecules, mRNA, various signalling molecules, and several other unknown secondary metabolites [1,40,47,135,136]. These studies raise the question of whether a similar function is played by inter- and intraxylary phloem. At present, there is no answer to this question and further studies on a similar line are needed, as carried out for members of Cucurbitaceae.

Besides intraxylary phloem, several members of the Amaranthaceae, Chenopodiaceae, Nyctaginaceae, Piperaceae, and *Argyreia nervosa* (Convolvulaceae) families show the presence of vascular bundles in the pith region, referred to as medullary bundles (Figure 13A–D). The vascular bundles that are positioned in the central ground tissue (pith) of the plants are arranged in one to two rings or scattered irregularly throughout the pith [22,50,137–139]. Their occurrence is not only reported in ferns but they are also recorded in nearly 60 families of flowering plants including magnoliids and eudicots [139]. Their presence has been observed in several species but due to the lack of exhaustive studies on this topic, it is little understood in several facets [139]. Therefore, in-depth studies are needed on their ontogeny, physiological significance, and evolutionary significance for a better understanding of their role. Each bundle is composed of a phloem and xylem with sandwiched cambial segments between them (Figure 13D). The development of medullary bundles initiates along with the primary growth stage (Figure 13D,E) and their number can vary from species to species (Figure 13A,B). The development of medullary bundles can take place concomitantly with the protoxylem and phloem or can initiate afterwards (Figure 13E). The number of medullary bundles can increase with the increase in age or remain constant. For instance, the number of medullary bundles increases with the increase in the age of *Argyreia nervosa* (Figures 13C and 14A), whereas the number remains constant in young and mature individuals of *Achyranthus aspera*, *Boerhavia*, *Cyathula*, *Celosia* (Figure 13B). After the initiation of the secondary growth stage, the segment of the cambium of these bundles divides and produces both types of conducting elements on their respective sides (Figure 13F). Functionally, the cambium is less active but the accumulation of collapsed and crushed phloem and increased number of xylem elements with wide vessels is an indication of the cell division activity of the cambium. In the Amaranthaceae, Chenopodiaceae, and Nyctaginaceae families, the xylem and phloem have a specific orientation, whereas in *A. nervosa*, they are randomly distributed in the pith and have an irregular orientation (Figure 14A). Medullary bundles initiate in the pith from the procambial stage and as the secondary growth stage progress, these bundles also undergo secondary growth and show the deposition of the secondary xylem and phloem, indicating it as an alternate pathway for the conduction of minerals, nutrients, and photosynthates, although there are other ways to achieve this function [50,58,109]. This statement can be supported by the presence of functional sieve elements adjacent to the cambium of the medullary bundles and the accumulation of a significant amount of collapsed and crushed phloem external to it.

### 3.3. Formation of Ray (Radial) Sieve Elements and Vascular Bundles in Rays

Ray sieve elements are the radially oriented sieve elements that differentiate from the ray cells in the phloem rays of some angiosperms [48,54,68,74,134,140]. They can be isolated or in a group and can occur in uniseriate to multiseriate rays. Dimensionally, they are like adjacent rays and can show the presence of a simple or compound sieve plate [54,74]. Their occurrence is a rare feature and one may not find it in every sample of the particular species investigated here. For instance, in *Sterculia urens*, *Tectona grandis*, *Erythrina indica*, etc. [54,74], all the rays do not show their presence. It is believed that hormones play an important role in the regulation of xylem and phloem differentiation [141]. The occurrence of radially oriented sieve elements in the rays of the secondary xylem and phloem has been reported in several species of phylogenetically unrelated families. It is not a conserved characteristic since its presence is observed occasionally and not in every ray. The members that show this occurrence belong to different habits (herbs, shrubs, trees, and climbers), indicating that there is no correlation with the habit. Uni- to multiseriate rays of *Barleria prionitis*, *Bombax ceiba*, *Erythrina indica*, *Ipomoea hederifolia*, *Leptadenia*, *Sterculia urense*, *Tectona grandis*, etc. (Figure 14B–F), show ray sieve elements. Species with interxylary phloem frequently show the development of such radial sieve elements (Figure 14C,D), which help to develop an anastomosing network for the short distance to connect to the adjacent phloem islands. These sieve elements can be isolated or in a group of a few cells. Since they develop from the ray cell initially, dimensionally, they are more or less similar to or

slightly larger than those of the adjacent ray cells (Figure 14D,F). The formation of such elements has a minimal effect on the adjacent cells since they are developed simultaneously with the adjacent ray cells. However, in some cases, such as *Adansonia digitata*, *Argyreia splendense, Hebanthe paniculata, Suaeda*, etc., the development of radial sieve elements is documented in response to the initiation of meristematic centres in already differentiated multiseriate rays (Figure 15A–C). Some of them (not all) show patchy meristematic activity due to the dedifferentiation of the ray cells of both the phloem (Figure 15E) and xylem (Figure 15D) sides, which initially deposit only sieve elements. Subsequently, some of the cells divide repeatedly to form small segments of the cambium (Figure 15D,E) that divide bidirectionally and produce radially oriented secondary xylem and phloem derivatives on either side of it. Therefore, the adjacent cells become compressed due to the addition of new derivatives of the secondary xylem and phloem and appear like radially arranged vascular bundles (Figure 15A–E).

Since the products of these vascular bundles develop from the meristematic cells, dimensionally, they differ from the isolated or group of sieve elements (Figure 15F). In Adansonia digitata, the length of these sieve elements is significantly more (285 μm $\pm$ 4.239) than the adjacent ray cells and they were smaller than the regular external (432 μm $\pm$ 3.988) sieve elements. In contrast, other species with these radial bundles show relatively shorter sieve elements, indicating that the variation in the dimensions of these products can vary from species to species. Experimental studies indicated that a low concentration of auxin induces only sieve element differentiation, whereas a higher concentration of these hormones is needed to produce xylem elements [142]. This may be the case with *Adansonia digitata* in which the deeply situated xylem rays in fully grown thick stems showed polycentric meristematic centres [48].

The sieve elements differentiating from these meristematic centres are significantly longer than the adjacent ray cells. They are radially elongated and sometimes even show the deposition of the xylem elements and appear like radially oriented vascular bundles. Lev-Yadun and Aloni [143] reported that these meristematic centres initiate in the large rays of the cambium due to repeated cell divisions in *Suaeda monoica*. However, the case of *A. digitata* differs from the *S. monoica* because the former species shows regular secondary growth and has a single ring of vascular cambium, whereas *Suaeda monoica* is characterized by the presence of successive cambia and other kinds of variant secondary growth [76,143]. Similar types of rays have also been described in *Hebanthe eriantha* [120] and *Argyreia splendense* [51]. Species with cambial variants frequently show the formation of the anastomosing network to interconnect the interxylary phloem or successive rings of the cambia and phloem [68,119,120]. It has a role in the rapid conduction of photosynthates in the radial direction but the differentiation of xylem derivatives such as tracheids and vessels indicates that they also play an additional role in the transport of water, minerals, and nutrients. The previously formed secondary xylem of *Adansonia* is characterized by the presence of abundant parenchyma cells that are thin-walled and unlignified. Deeply situated parenchyma (nearly 25–30 cm away from the cambium) show their proliferation and consequently form unusual tissue complexes [48]. The repeated cell division and differentiation of vascular tissues in the locally induced areas may demand a higher concentration of photosynthates. It is a well-known fact that sieve tube elements are associated with photosynthate translocation, whereas tracheary elements play a vital role in the transport of water, minerals, and nutrients, and the only difference is that their orientation is radial instead of axial.

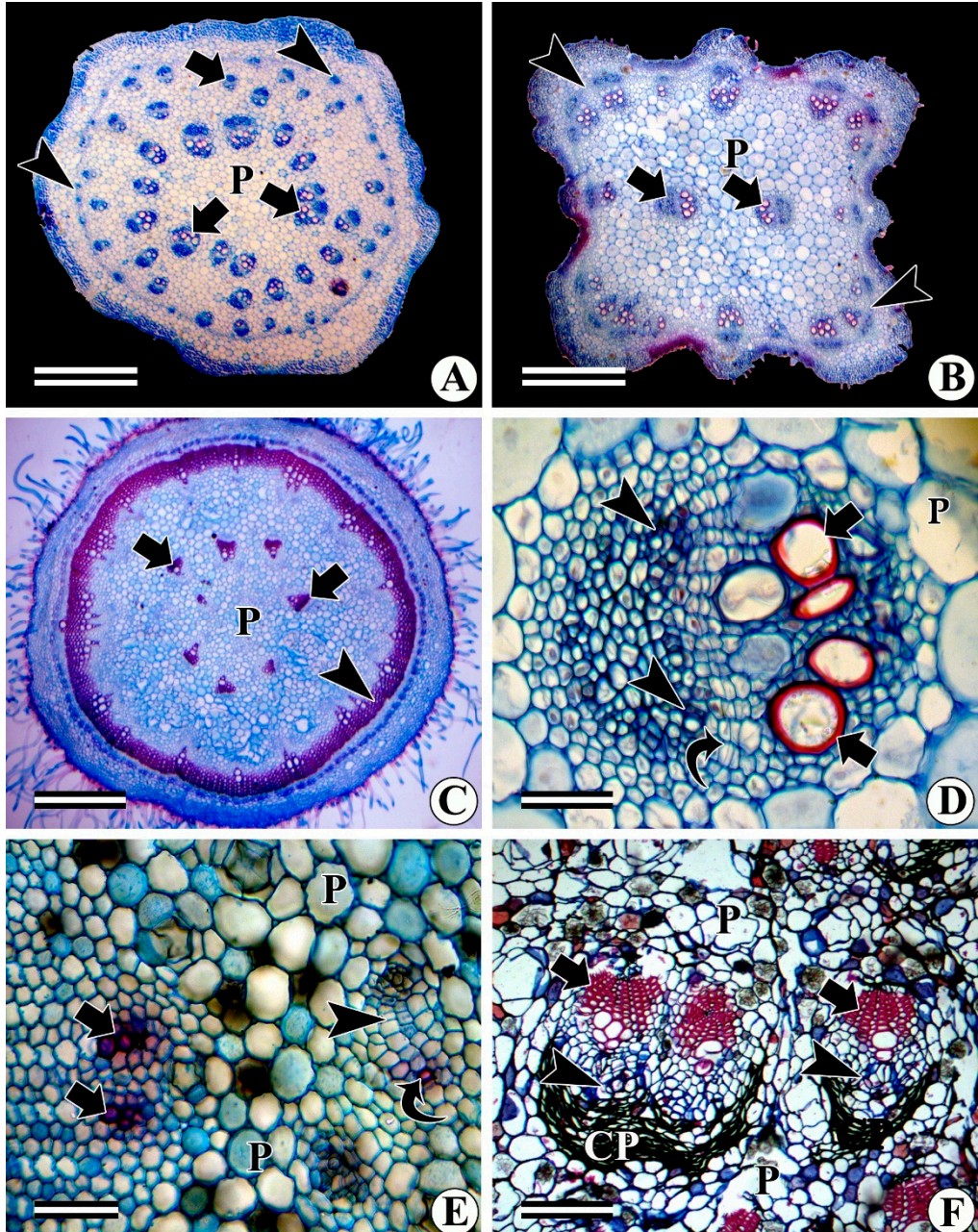

**Figure 13.** Transverse view of young and mature stems showing medullary bundles in different species of eudicots. (**A**) Young stem of *Amaranthus spinosus* showing several medullary bundles (arrows) in the pith region. Note the initiation of the vascular cambial ring (arrowheads). (**B**) Young stem of *Achyranthes aspera* showing only two medullary bundles (arrows). Arrowheads indicate initiation of the regular vascular cambium. (**C**) Young stem of *Argyreia nervosa* showing a few medullary bundles (arrows). Arrowhead shows a well-established ring of secondary xylem. (**D**) Enlarged view of one of the medullary bundles in *Achyranthes aspera* showing the structure and composition of the medullary bundle. Note the phloem (arrowhead), protoxylem (arrow), and procambium (curved arrow). (**E**) Simultaneous origin of protoxylem (arrows) and the medullary bundles (curved arrow) in *Argyreia nervosa*. Arrowhead indicates the cambial segment of the medullary bundle. Note the absence of intraxylary protophloem between the protoxylem and the medullary bundles. (**F**) Secondary growth in the medullary bundles of *A. nervosa* showing a quantifiable amount of secondary xylem (arrows) and phloem (arrowheads). Note the crushed phloem. *Abbreviations*: CP = crushed phloem, P = pith. ***Scale bars***: (**A–C**) = 500 μm, (**D,E**) = 50 μm, (**F**) = 200 μm.

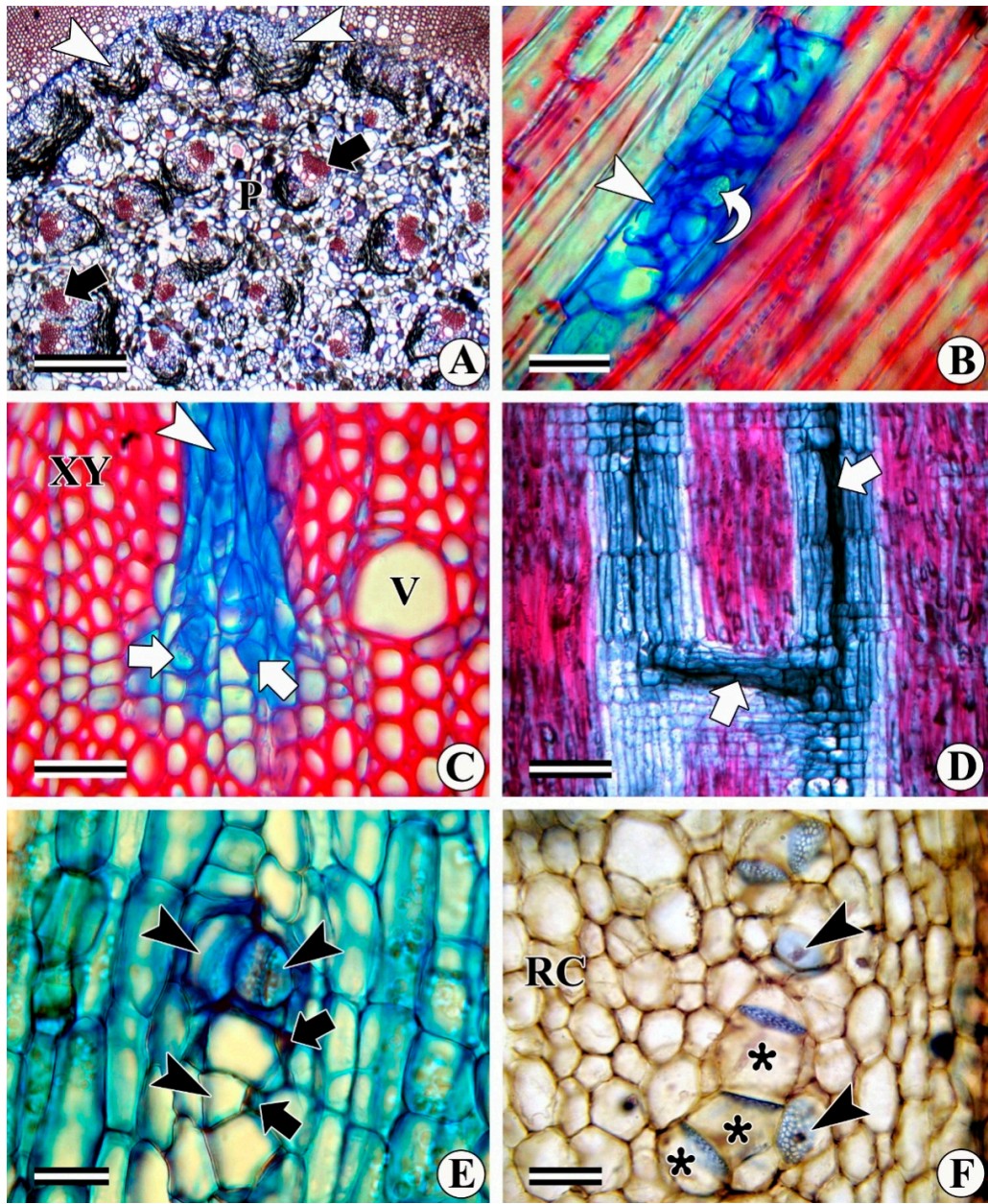

**Figure 14.** Fully grown mature stems showing medullary bundles (**A**) and ray sieve elements in different eudicots. (**A**) Fully grown mature plants showing an increased number of medullary bundles (arrows) in *Argyreia nervosa*. Arrowheads indicate intraxylary phloem, which was absent in the young stem (pl. see Figure 13C). (**B**) Tangential longitudinal (TLS) view of *Leptadenia pyrotechnica* secondary xylem showing radially oriented sieve element (curved arrow) in the xylem ray (arrowhead). (**C**) Transverse views of *L. pyrotechnica* stem showing sieve plates (arrows) in the xylem rays. Arrowhead indicates radially elongated ray sieve elements. (**D**) Adjacent interxylary phloem (arrows) showing anastomosing in *Salvadora oleoides.* (**E**) TLS view of *Ipomoea hederifolia* showing ray sieve element (arrowheads), with the arrows showing the companion cells. (**F**) TLS view of *Erythrina indica* showing ray sieve elements (arrowheads). Note the orientation of sieve elements shown with arrowheads and asterisks. *Abbreviations*: P = pith, RC = ray cells, V = vessel, XY = secondary xylem. ***Scale bars***: (**A**) = 500 μm, B, (**D**–**F**) = 50 μm, (**C**) = 100 μm.

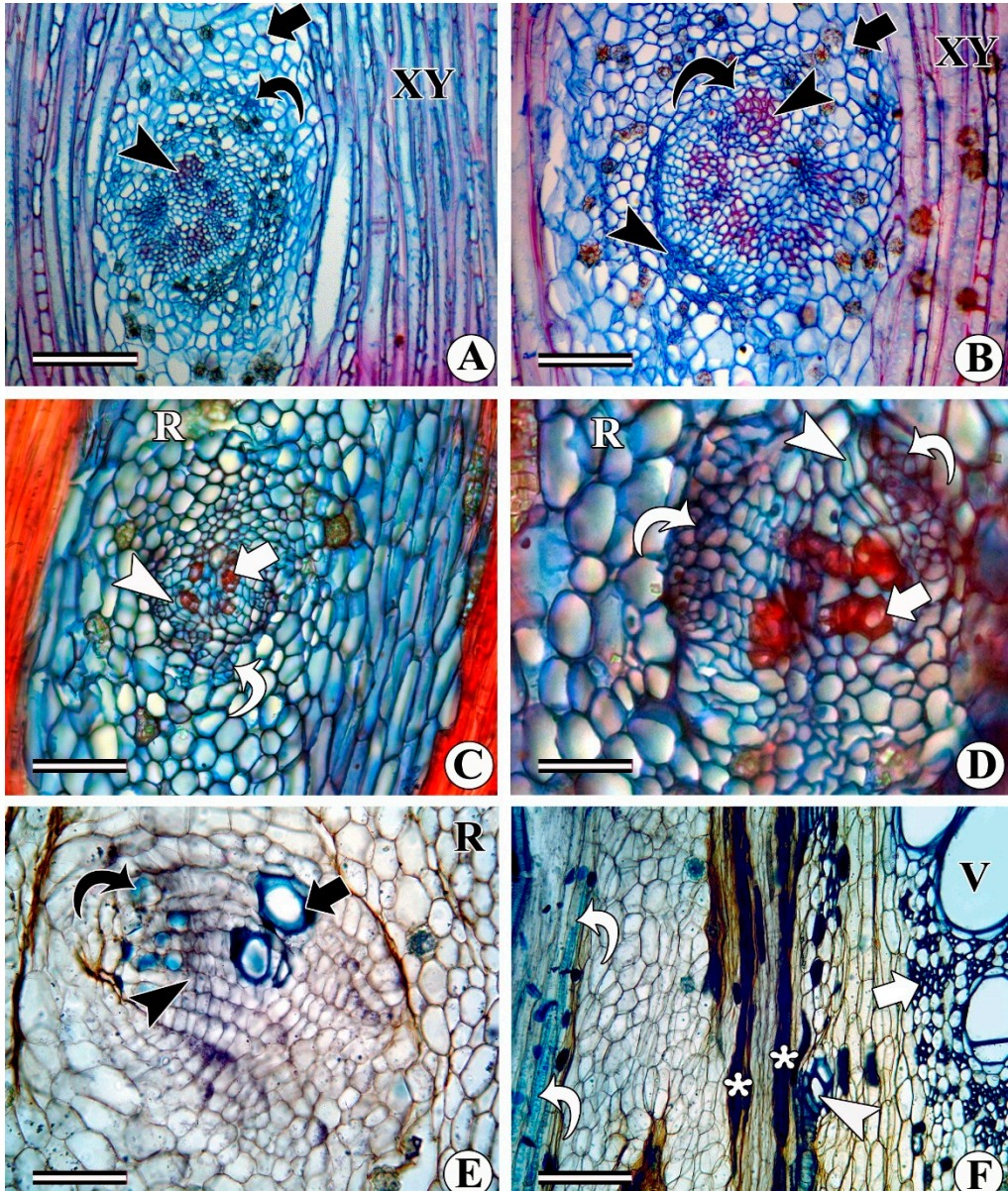

**Figure 15.** Tangential longitudinal (**A–E**) and transverse (**F**) views of the mature stems of different eudicots showing sieve elements in the xylem and phloem rays. (**A**) Multiseriate xylem rays (arrow) in *Argyreia splendense* showing meristematic centres with xylem (arrowhead) and phloem (curved arrow). (**B**) Magnified view of the multiseriate rays (arrow) in *A. splendense* showing meristematic activity in ray cells (curved arrow) that produced xylem and phloem (arrowheads). (**C**) Meristematic centres in one of the multiseriate rays in *Hebanthe paniculata* showing meristematic cells (arrowhead) and their products, i.e., xylem (arrow) and phloem (curved arrow). (**D**) Enlarged view of meristematic centre in one of the multiseriate rays in *H. paniculata*. Figure legend follows a similar indication. (**E**) Tangential longitudinal view of secondary phloem ray of *Adansonia digitata* showing meristematic centre in multiseriate ray. Arrowhead shows meristematic cells, phloem (curved arrow), and vessel element (arrow). (**F**) Meristematic centre in a multiseriate ray of *A. digitata* showing radial sieve elements (curved arrows) and vessels (arrowhead) and the asterisks indicate crushed phloem. Note the vessel (V) and fibres (arrow) in the transverse view of the xylem. *Abbreviations*: R = ray, V = vessel, XY = secondary xylem. ***Scale bars***: (**A**) = 200 µm; (**B,C,F**) = 100 µm; (**D,E**) = 50 µm.

*3.4. Formation of Phloem Wedges*

It is a well-known fact that the liana habit evolved with several alterations in their mechanical architecture to increase their stem flexibility during stem torsion [9,81,82,97]. During this shift from self-supporting to non-self-supporting habits, there were significant alterations in the mechanical tissues [9,49,74,81,83–87]. In most eudicots, cambium forms lignified elements of the secondary xylem centripetally and phloem centrifugally irrespective of whether the stem conformation is circular, lobbed, flattened, or variously shaped. Some species, particularly in plants with climbing habits, show the formation of phloem wedges, which is one of the alterations in which a discrete segment of the vascular cambium either ceases to produce the secondary xylem or produces less xylem and shows an elevated phloem development. Such differential activity of the vascular cambium results in the establishment of depressions that eventually become deep phloem wedges. The formation of such phloem wedges is characteristic of a climbing habit and has been reported in families such as Bignoniaceae, Malpighiaceae, Convolvulaceae, etc. [9,71,144–146]. Phloem wedges, also referred to as furrowed xylem, are one of the prevalent cambial variants that have been observed in several members of flowering plants. Their occurrence has mostly been reported in climbing species and, to some extent, they remain characteristic of a specific family. For example, the development of phloem wedges is typical in the tribe Bignonieae of the family Bignoniaceae, which develops four (Figure 16A,B) or multiples of four furrows in different taxa. *Serjania mexicana* shows a similar type of phloem wedge. However, it shows a few variations from those of the tribe Bignonieae in terms of the number of furrows/wedges (Figure 16C). In contrast, the development of phloem wedges differs in Malpighiaceae (Figure 16D) and some members of the Convolvulaceae family such as *Hewittia, Jacquemontia, Merremia,* etc. (Figure 16E). The number of wedges increases with the upsurge in the age of the species. In contrast, *Neuropeltis racemosa* (Convolvulaceae) differs from all the above-mentioned types (Figure 16F).

Ontogenetically, the aforementioned types of phloem wedges vary from each other. In the members of the tribe Bignonieae, the formation of wedges initiates after a short period of regular secondary growth. In the tribe Bignonieae, four segments of the cambium alter the cell division activity equidistantly. These segments become functionally less active/cease to divide centripetally and increase the phloem development. In contrast, alternating adjacent segments of the cambium, i.e., interwedges divide regularly and deposit secondary xylem and phloem. Therefore, it forms four curved arcs/depressions of the cambium (Figure 16B). In some of the species, the four arcs show the increased activity of the cambium and continued phloem deposition by retaining the cambium as a continuous ring, whereas in other species, it forms four disjunct segments of the cambium due to the limiting rays. With the increase in age, depending on the species, the number of wedges can increase in multiples of four by forming additional wedges in the interwedge portion of the cambium. However, in *Dolichandra unguis-cati* (also referred to as cat's claw) the formation of phloem wedges differs from the other members of the tribe Bignonieae (Figure 17A). This species has a large portion of the unlignified parenchyma and limiting rays. Over time, these unlignified cells become meristematic and divide into all planes. Therefore, not only are these phloem wedges compressed but they also become dissected and are enclosed within the secondary xylem. Such behaviour results in a mosaic of phloem wedges and includes the phloem islands (Figure 17A) that were embedded within the secondary xylem. Family Bignoniaceae, especially the tribe Bignonieae is a well-known example of the formation of such wedges [22,56,77,100,105,147]. The occurrence of phloem wedges was reported for the first time by De Jussieu [93], whereas Schenck [77] provided detailed information on their development c.f. [81]. The formation of phloem wedges in *Dolichandra unguis-catis* (syn. *Doxantha unguis-catis*) was described in detail by Dobbins [105]. In the thick stem of this species, it becomes much more intricate due to the development of additional phloem wedges and the proliferation of parenchyma cells. The formation of additional phloem wedges in the interwedge region of the cambium and the proliferation of the xylem parenchyma was documented by Angyalossy et al. [96]. A similar case occurs with

the formation of furrowed xylem in *Neuropeltis* [10,81] in which the formation of phloem wedges is associated with the proliferation of parenchyma cells and differentiation of xylem derivatives that leads to the embedding of these wedges within the secondary xylem.

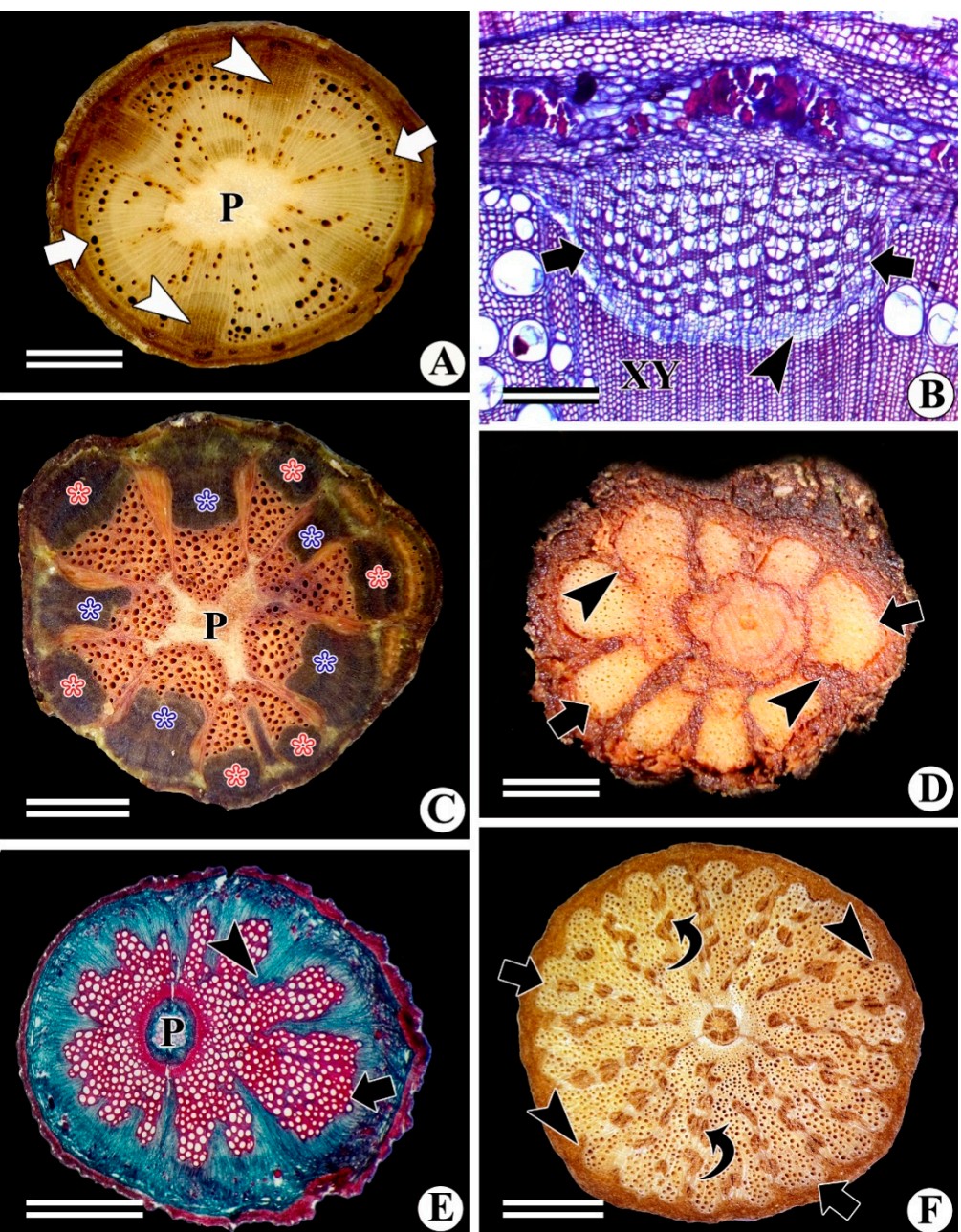

**Figure 16.** Transverse view of macro-morphology of young and mature stems (**A**,**C**–**F**) and section (**B**) of various taxa showing phloem wedges. (**A**) The young stem of *Mansoa alliacea* (syn. *Bignonia alliacea*) shows four phloem wedges (arrowheads), with arrows indicating interwedges. (**B**) Portion of phloem wedge (arrows) in *Arrabidaea candicans* showing the variant cambium that results in the phloem wedge (arrowhead). (**C**) Thick stem of *Serjania mexicana* showing phloem wedges (asterisks). Initially formed phloem wedges with blue asterisks and later formed phloem wedges shown with red asterisks. (**D**,**E**) Mature stem of *Diplopterys carvalhoi* (**D**), and *Jacquemontia pentanthos* (**E**) showing phloem wedges (arrowheads), with arrows indicating interwedges. (**F**) The mature stem of *Neuropeltis racemosa* showing phloem wedges (arrowheads) and interwedges (arrows). The curved arrows indicate wedges enclosed within the secondary xylem. *Abbreviations*: P = pith, XY Secondary xylem. ***Scale bars***: (**A**) = 2 mm; (**B**) = 200 μm; (**C**,**E**,**F**) = 5 mm; (**D**) = 7 mm.

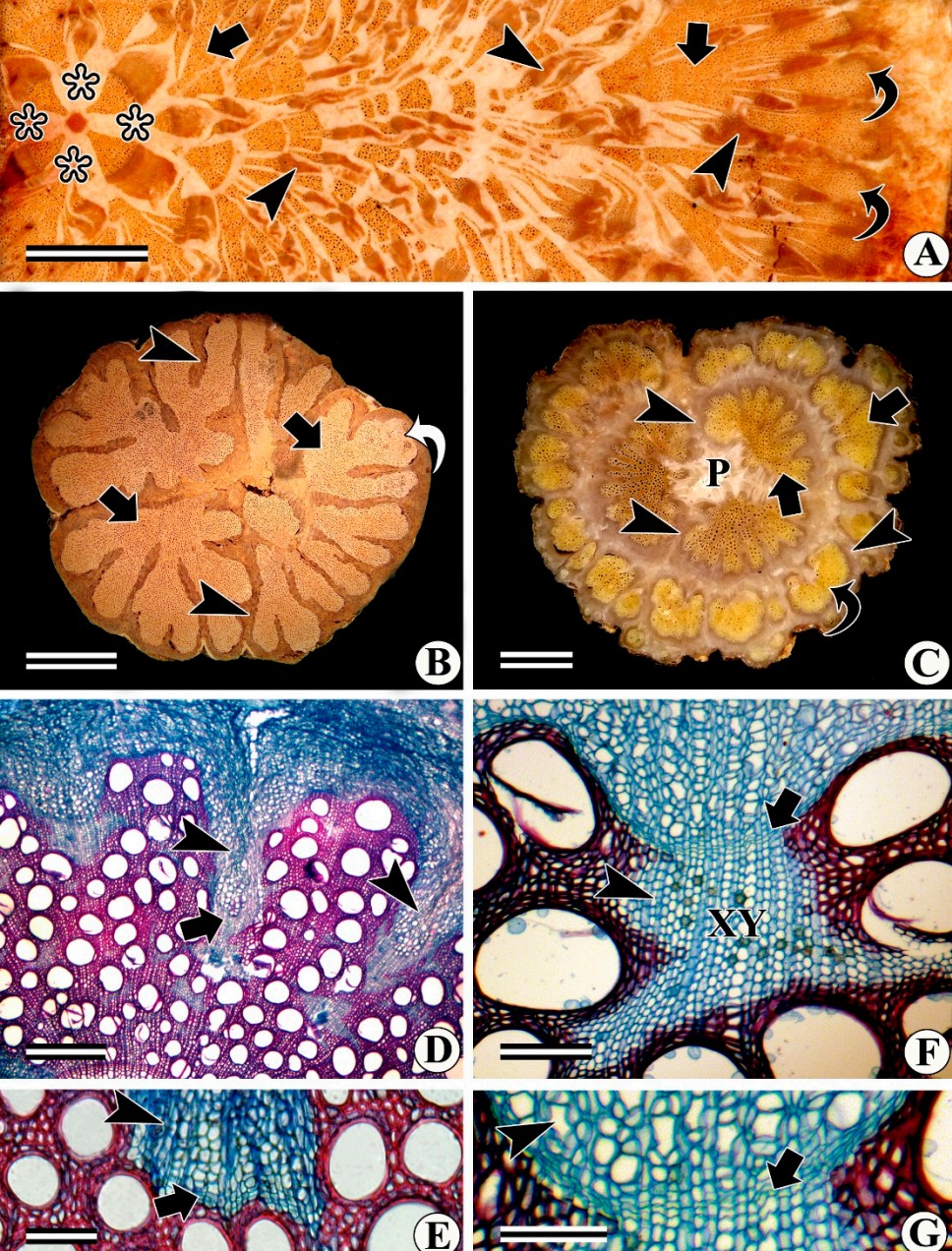

**Figure 17.** Macro-morphology of thick stems (**A**–**C**) and transverse view of the stem (**D**–**G**) showing phloem wedges in different eudicots. (**A**) Macroscopic view of *Dolichandra unguis-cati* showing the structure of the secondary xylem. Note the initial four wedges (asterisks), xylem formed in interwedges (arrows), phloem (arrowheads), and interwedges (curved arrows). (**B**,**C**) Mature stem of *Heteropterys* sp. (**B**) and *Hewittia malabarica* (**C**) showing phloem wedges (arrowheads), xylem formed by the cambium of interwedges (arrows), and interwedges (curved arrow). (**D**) Phloem wedges in *Jacquemontia pentanthos* showing phloem wedges (arrowheads). Note the position of the cambium (arrow). (**E**) Enlarged view of Figure 17D showing phloem wedge (arrowhead) and position of the cambium (arrow). (**F**) Phloem wedges in *Merremia vitaefolia* showing phloem wedges (arrowhead). Note the position of the cambium (arrow). (**G**) Enlarged view of Figure 17F showing phloem wedge (arrowhead) and position of the cambium (arrow). *Abbreviations*: P = pith, XY = xylem. ***Scale bars***: (**A**) = 7 mm; (**B**,**C**) = 8 mm; (**D**) = 500 μm; (**F**) = 200 μm; (**E**,**G**) = 100 μm.

*Serjania mexicana* (Sapindaceae) shows a similar type of cambial variant, although it follows a similar pattern of phloem wedge development as that observed in tribe Bignon-

ieae; however, it differs in that it has an odd number of phloem wedges (i.e., 3 or 5) and this number increases with the upsurge in the age of the plants (Figure 17A). In contrast, the development of the phloem wedges differs from the above-mentioned types in Malpighiaceae and some members of the Convolvulaceae family such as *Hewittia, Jacquemontia, Merremia*, etc., (Figure 16D,E). In *Diplopterys carvalhoi, Heteropterys* sp. (Malpighiaceae), and some members of the Convolvulaceae family, the phloem wedges show few variations in their development. In *D. carvalhoi* and *Heteropterys*, initially, the vascular cambium is functionally regular and maintains a circular outline of the stem/vascular cambium by depositing the secondary xylem and phloem equally across the stem. Soon after, certain regions of the cambium modify their cell division activity and begin to deposit more secondary phloem and less xylem. Such behaviour of the vascular cambium consequently forms shallow invagination/arcs, resulting in the formation of phloem wedges. As the stem thickness increases, the wedges become deeper and deeper (Figure 17B) and can even lead to the formation of lobbed/grooved stems (Figure 17C). However, in Convolvulaceae, the formation of phloem wedges follows two different methods of phloem wedge development based on their ontogeny and time of initiation. Taxon such as *Hewittia malabarica, I. pescaprae, Merremia hederacea*, and *M. vitaefolia* show both types of wedge formation, whereas *Jacquemontia* (Figure 17D) follows a similar pattern to that of the family Malpighiaceae. The phloem wedges that develop only after the formation of a quantifiable amount of secondary xylem (Figure 17D,E) show the cessation of cell division by the vascular cambium towards the secondary xylem, which produces only phloem elements externally (Figure 17D). In the second type, the phloem wedges begin to form immediately after the initiation of the secondary growth stage in which the cambium produces unlignified parenchyma internally and phloem externally to maintain a circular outline or it forms a shallow invagination (Figure 17F,G) due to the relatively fewer cell division activities towards the xylem side. *Jacquemontia pentantha* and *Merremia vitaefolia* also show the development of phloem wedges by maintaining a continuous cambial ring. However, it deviates slightly in that the cambium becomes functionally slow and deposits unlignified parenchyma internally and more phloem externally. With time, these parenchyma cells dedifferentiate into sieve tube elements, as reported for other members of the Convolvulaceae family [49,56,60,62,74,130]. Here, the only difference is that its development takes place in the xylem parenchyma cells that are produced by the cambium, which is responsible for the formation of the phloem wedge. Subsequently, phloem wedges have also been reported in families such as Asteraceae, Malpighiaceae, and Sapindaceae [9,71,144–146].

## 4. Interxylary Phloem in Some Families of Caryophyllales

Caryophyllales have long been a topic of discussion due to their unique and atypical cambial activity. There is a difference of opinion regarding the status of phloem since it remains embedded in conjunctive tissues instead of the secondary xylem. Therefore, herewith, it is treated as a special case of interxylary phloem development. The formation of interxylary phloem islands in some of the species we investigated and the fate of the phloem islands that were developed a few years before in perennial species such as *Arthrocnemum indicum, Suaeda, Pupalia, Boerhavia, Bougainvillea,* etc., is subsequently elucidated. Like other woody plants, in members of the Caryophyllales family, secondary growth initiates regularly and produces secondary xylem centripetally and secondary phloem centrifugally (Figure 18A). However, due to the differential activity of the vascular cambium, a small portion produces conducting elements, whereas alternating segments exclusively form fibres like thick-walled, lignified sclerenchyma cells (Figure 18A), in which most of the fibres are characterized by the presence of a nucleus. In the early stage of secondary growth, a small segment of the cambium ceases to form the secondary xylem internally, whereas an adjacent portion of the cambium continues regular secondary growth. Therefore, it develops phloem islands from the small segments of coalescent cambium (Figure 18B–D), which form a complete ring of the vascular cambium. This newly formed coalescent cambial segment deposits the secondary xylem internally and the phloem externally. Therefore,

the phloem islands, along with a ceased segment of the cambium, are enclosed within the secondary xylem (Figure 18E,F). As the plant grows, the tangential width of such coalescent cambial segments increases gradually (Figure 18E) and ultimately, in thick stems, a complete ring of the cambium is replaced by the formation of a complete ring of the successive ring of the cambium (Figure 18F). Similar results were also documented by Cunha Neto et al. [121], who reported the formation of similar coalescent cambia and successive cambia in different species of Nyctaginaceae.

*Fate of Previously Formed Phloem Islands*

The development of interxylary phloem has been investigated worldwide from various angles by several researchers [76,120,122]. However, what happens to the interxylary phloem islands that were formed immediately after the initiation of secondary growth in perennial species? This is the most neglected aspect and there is a lack of related information available in the literature. The investigation of a few species shows that these islands function for a considerable period and replace the non-conducting elements of the phloem by depositing new sieve elements. Irrespective of the families of Caryophyllales, some of the perennial species with interxylary phloem islands (*Arthrocnemum indicum*, *Bougainvillea* sp., *Gallesia integrifolia*, *Suaeda monoica*, *Salvadora oleoides*, *Strychnos bredemeyeri*, *S. potatorum*, etc.) undergo a gradual replacement of phloem derivatives and unlignified parenchymatous cells with lignified elements and finally, such islands are completely replaced with lignified elements. As mentioned earlier, each island is accompanied by a small segment of cambium that is functionally less active. The addition of new derivatives gradually replaces the non-conducting sieve elements either with the addition of new sieve elements or by forming associated parenchyma cells. Over time, the fusiform cambial cells of the previously formed phloem islands gradually lose their radial arrangement (Figure 19A–C). After the loss of the radial arrangement of the cambial cells, the associated phloem parenchyma cells of these islands enlarge and differentiate into xylem or phloem derivatives concomitantly with the collapse of the non-conducting sieve elements (Figure 19D–F). Progressively more and more phloem parenchyma cells are differentiated into lignified elements/fibres or vessel elements. Repeating this action ultimately reduces the size of the phloem island and eventually, the entire phloem island is sealed and occupied by the lignified derivatives, whereas the crushed phloem of these islands remains restricted to the small cavity or the cell corner (Figure 19F).

As shown in Figure 19C–F, the non-conducting and crushed phloem is subsequently replaced and sealed due to the lignification of the last surviving cells. We presume that in thick stems, these phloem islands are deprived of the growth hormones, minerals, nutrients, gaseous exchange, etc., which are required for survival. Therefore, this can be hypothesized and may be comparable with heartwood formation.

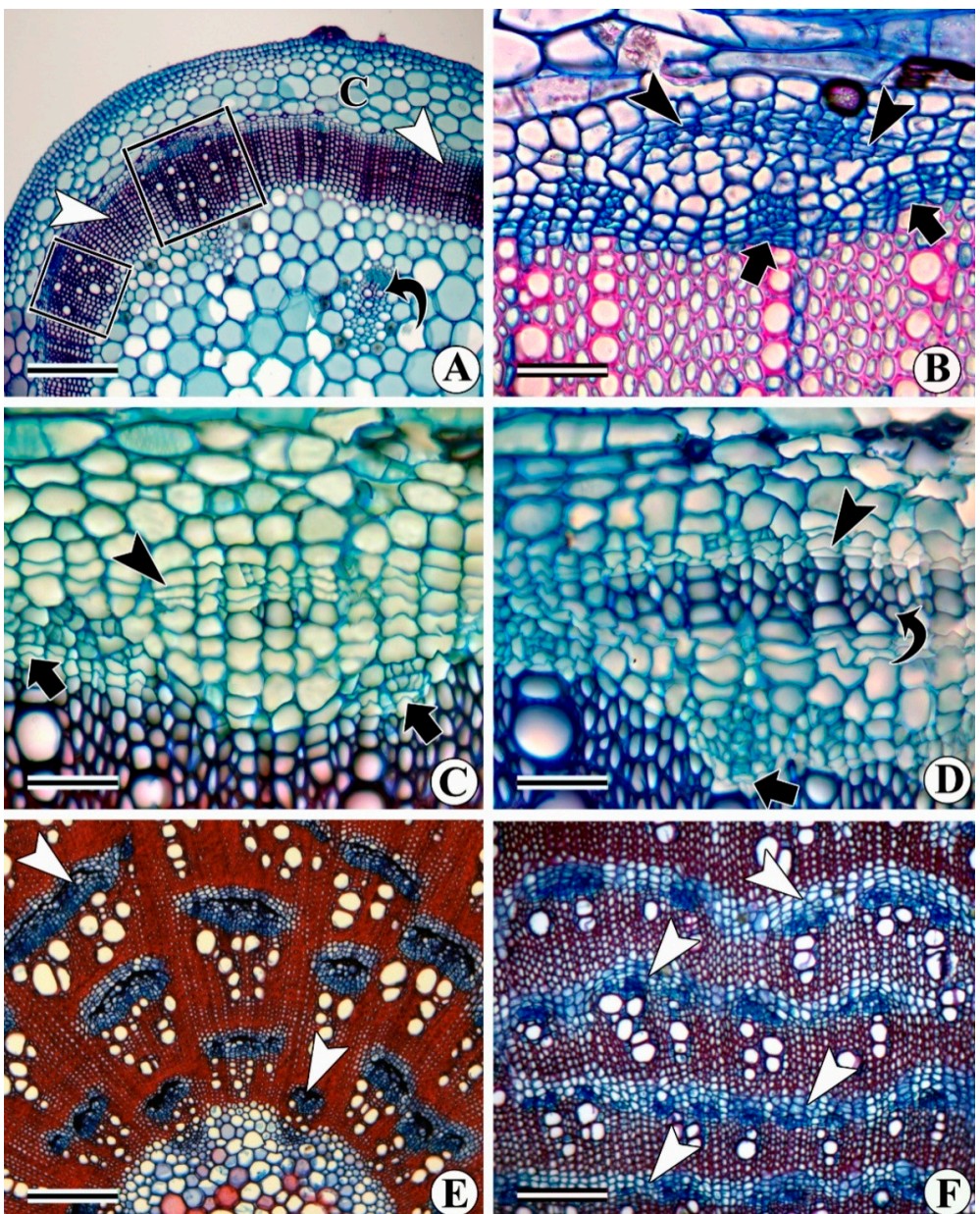

**Figure 18.** Transverse view of stem showing various stages of phloem development in different members of the Caryophyllales family. (**A**) Establishment of the regular vascular cambium in *Cyathula prostrata* showing a completely formed xylem ring. Differentiation of the xylem and phloem is restricted to a certain portion of the cambium (rectangles), whereas alternate segments show only the development of fibres (arrowheads). The curved arrow indicates the medullary bundle. (**B–D**) Development of coalescent cambium (arrowhead); arrow(s) indicate the ceased portion of the cambium in *Nothosaerva brachiata*, *Allmania nodiflora,* and *Alternanthera sessilis,* respectively. The curved arrow in Figure 18D indicates the products of the newly formed cambial segment. (**E**) Thick stem of *Hebanthe paniculata* showing previously formed phloem islands and coalescent phloem parenchyma (arrowheads) embedded within the lignified elements. Note that the formation of the xylem and phloem is restricted to a certain portion of the cambium and the gradual increase in the tangential width of the coalescent cambium. (**F**) Complete successive ring (arrowheads) formation externally in thick stems of *Aerva sanguinolenta*. **Scale bars**: (**A,E**) = 200 μm; (**B–D**) = 50 μm; (**F**) = 100 μm.

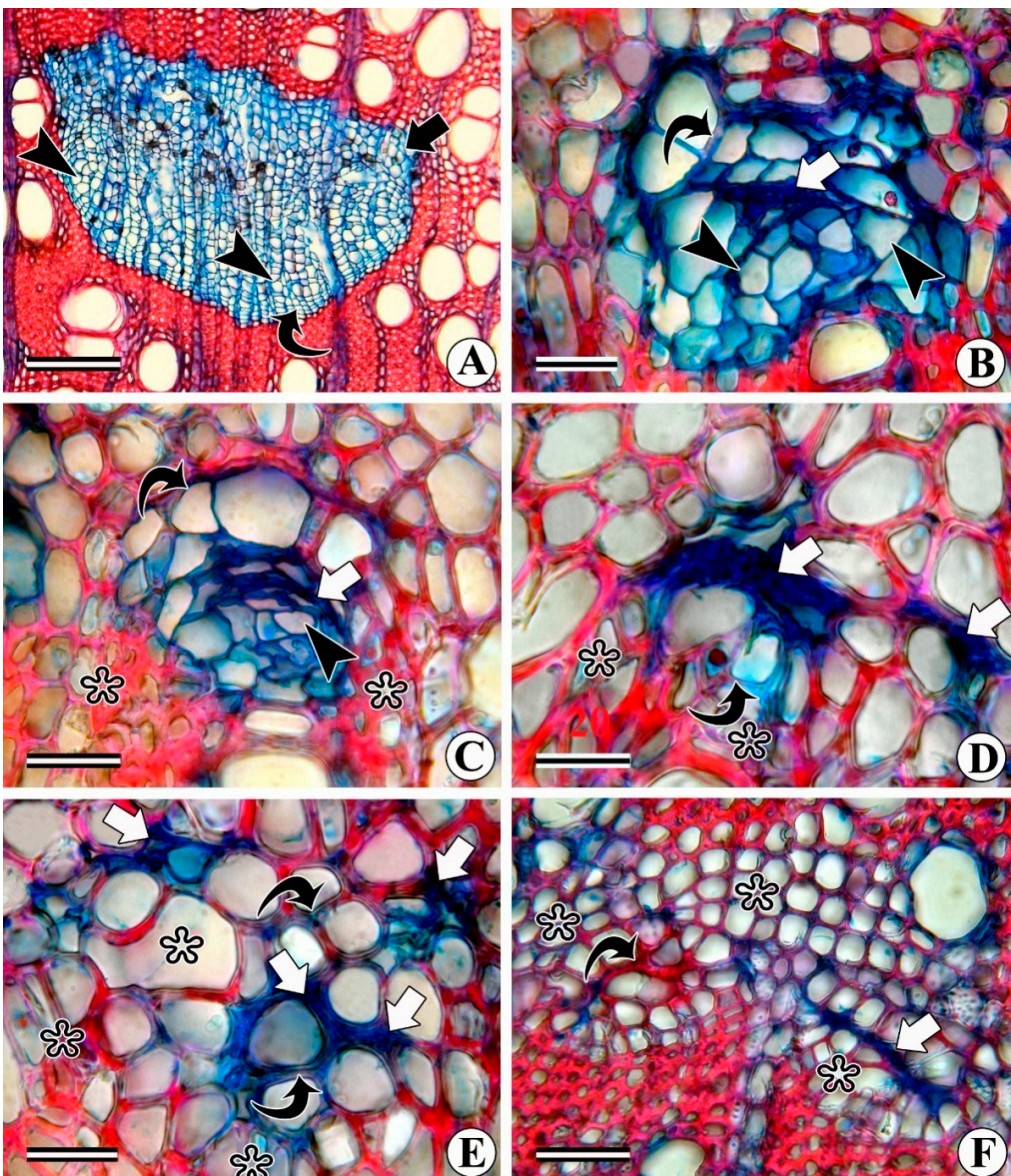

**Figure 19.** Transverse view of interxylary phloem islands formed at the beginning of the secondary growth stage in different species of eudicots. (**A**) Interxylary phloem island in *Strychnos bredemeyeri* (arrow). Arrowheads indicate phloem and curved arrow shows cambium. (**B**) One of the previously formed phloem islands in *Gallesia integrifolia* showing crushed phloem (arrow), sieve elements (arrowheads), and partly lignified cell wall of the adjacent parenchyma cells (curved arrow). Note the loss of radial arrangement of the cambial cells. (**C–E**) Gradual reduction in the size of the interxylary phloem island. Note the crushed phloem (arrow/s) and partly lignified parenchymatous cells (curved arrow). Note the irregular-sized cells (asterisks) that have undergone lignification. (**F**) Completely sealed interxylary phloem island (curved arrow); arrow indicates partially sealed island (note the astra blue staining) and adjacent parenchyma cells (asterisk) that have undergone lignification (curved arrow). *Scale bars*: (**A**) = 100 μm; (**B–E**) = 20 μm, (**F**) = 50 μm.

## 5. Conclusions

The origin of the vascular tissues during evolution played a vital role in the emergence of diverse habits in vascular plants for supplying minerals, nutrients, photosynthates, and several signalling molecules. A growth adjustment for survival led to alterations in the structure, composition, and allocation of the different conducting elements of the xylem and phloem that were not in their regular positions. Specifically, plants with climbing habits

(including some self-supporting species such as *Gallesia, Phytolacca, Strychnos, Suaeda,* etc.) are characterized by the presence of successive rings of the secondary xylem and phloem. Some of them develop islands of phloem enclosed within the secondary xylem (interxylary phloem), whereas others show the presence of phloem in the pith region (intraxylary phloem) or form complete vascular bundles (medullary bundles). The development of radially oriented sieve elements is not a consistent feature and it is a case of autapomorphy. Therefore, the phloem forms a cable/wiring network that connects the distantly separated organs by forming an anastomosing network that can help with the exchange of minerals, nutrients, photosynthates, and signalling molecules. The formation of inter- and intraxylary phloem differs ontogenetically and topographically from species to species. Primary intraxylary phloem can develop simultaneously, before, or after the formation of regular protoxylem and protophloem. Depending on the species, additional secondary intraxylary phloem is deposited from the adjacent pith cells. With an increase in age, several species develop intraxylary phloem cambium that can produce only phloem centripetally, whereas the bidirectional differentiation of the secondary xylem is a rare feature. A lack of experimental evidence (except on regular phloem) warrants further studies on the role of a phloem that is not in its regular position. The formation of such a phloem is ascribed to the rapid translocation of photosynthates from source to sink. Experimental evidence on the members of Cucurbitaceae shows that in addition to photosynthates, extra-fascicular phloem plays important role in the conduction of other macromolecules. Similar studies are needed on inter- and intraxylary phloem in a greater number of plant species.

**Author Contributions:** Conceptualization, investigation, project administration; funding acquisition, supervision, and writing—review and editing, K.S.R.; collection of plant material, methodology, writing—original draft preparation; formatting, K.K.K., D.G.R., K.D.T. and A.D.G. All authors have read and agreed to the published version of the manuscript.

**Funding:** This research was funded by the Science and Engineering Research Board (SERB-DST, Grant No. SR/SO/PS-179/2012 dated 06/08/2013), University Grants Commission (UGC, Grant No. 33-195/2007 (SR) dated 13/03/2008 and No. F.3-48/2003 (SR) dated 30/03/2003), Council of Scientific and Industrial Research (CSIR, Grant No. 38(1289)/11/EMR-II dated 26/04/2011), Government of India. The authors would like to acknowledge Gujarat Biodiversity Board (No. GBB/Survey Studies/910-14/2013-14), Gujarat State Biotechnology Mission (No. GSBTM/MD/PROJECTS/SSA/4892/2015-16) Govt. of Gujarat for the financial support to carry out the work.

**Data Availability Statement:** Not applicable.

**Acknowledgments:** The authors are thankful to the Department of Forests, the Government of India, and the Officer In-charge of respective forest areas of Gujarat and Maharashtra states for their wholehearted support during the fieldwork. One of the authors (KSR) is also thankful to Carmen Marcati (Departamento de Recursos Naturais, Faculdade de Ciências Agronômicas, UNESP, Campus de Botucatu, SP, CP237, CEP18603-970, Brazil) and Roger Moya (Escuela de Ingeniería Forestal, Instituto Tecnológico de Costa Rica, Cartago, Costa Rica) for their support during the stay. KSR is highly indebted to CAPES (Ministry of Education, Brazil), CTC (Costa Rica), and the Maharaja Sayajirao University of Baroda, Vadodara (India) for the financial support and necessary permissions to carry out the work in Brazil and Costa Rica. Authors are also thankful for the Oxford Academics; Taylor and Francis and Società Botanica Italiana; Royal Botanical Society of Belgium; Torrey Botanical Society; Universität Graz; Polish Academy of Sciences, Kraków; Faculty of Biology, University of Murcia, for necessary permission to reuse some images.

**Conflicts of Interest:** The authors declare no conflict of interest.

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
