# Peer review of "Inter- and Intraxylary Phloem in Vascular Plants: A Review of Subtypes, Occurrences, and Development"

_forests, doi:10.3390/f13122174_

Round 1

Reviewer 1 Report

In this article, Kishore Rajput and collaborators revise all the types of phloems in different positions other than the most common one (around a xylem core) is found. On my opinion the MS is well-written, interesting and a sound contribution to the field. I also congratulate the authors for the superb figures, that are fundamental to the article. I recommend the article for publications and I believe it is an excellent fit for the special edition being organized by Forests. 

The have some caveats that need addressing, though

The first one regards the title. It doesn't reflect the content of the article, because there isn't anything on phloem regulation, and also not all phloems treated in this article are within xylem. I suggest something like: Inter and intraxylar phloem in plants: subtypes, occurrence and development. 

or yet: Phloems in odd positions: a review of inter and intraxylary phloem in vascular plants. 

Also, I urge the authors to eliminate completely citations of the term “dicotyledons” , since it has been dropped decades ago. Substitute for “woody angiosperms” or "magnoliids and eudicots" like they rightfully did in some sections. Do the same throughout the text, whenever “dicots”are mentioned. 

Abstract: Make it more clear that this is a review article, because at first I was confused trying to identify what is background and what was the result of this work. Only after reading the entire MS I understood this is a review paper. Make it clear from start. 

Eliminate “higher plants”, just use “vascular plants” or “euphyllophytes” if wanting to have the licophytes excluded, or “seed plants” if wanting to have only gymnosperms and angiosperms. The term is used also in other parts of the MS and needs to be eliminated, since it was also dropped long ago.

I disagree with the following proposition that “The occurrence of such phloem is characteristic to certain families or groups of plants that either have a characteristic habit or habitat.” Intraxylary phloem is ubiquitous in Myrtales, where plants of all habitats and plant habits are to be found. 

What is “successive xylary phloem cambia”. The term commonly seen in the literature is “successive cambia”. The cambium is neither xylary or phloem, so this would be a misnomer.  

Introduction

What are “developmental provocations”? This sentence is an almost identical sentence from the cited paper “Rodríguez-Villalon” only by substituting some words, such as enables for empowers and stimuli for provocation. But provocations reads incorrectly here. I suggest completely rephrasing this sentence.

The authors say sieve cells are a synonym of Strasburger cell. That is a conceptual mistake, since they are completely different cell types. The Strasburger cells are parenchymatic cells associated to the sieve cells, which maintain these cells alive and functioning. 

Another conceptual mistake is calling the vascular system of the root a bundle. It’s not a bundle, because it is a protostele, therefore, it is a complete cylinder, and not a collection of bundles.

I disagree that not all contemporary scientists nowadays use phloem as a term. The authors cite a paper from Esau 1939 to support this and the use of other terms such as criblé in French or Cribalheit in German, but these terms are no longer recommended.  

Eliminate “higher plants” and “ lower plants”, these terms have been abolished for decades. Phaeophycean algae are no longer considered plants. 

The citation of Cucurbita in the context of this MS is debatable. Because the authors are talking about intra and interxylary phloem, but the article cited, from Turgeon and Oparka, is actually dealing with a third type of phloem present in the Cucurbitaceae, which they called interfascicular (in between the vascular bundles). It leads the readers to think the paper is talking about the internal phloem (intraxylary phloem). 

The aims are: “The present work aims to compile information on phloem at unusual phloem, their ontogeny and possible role based on our own work by using supporting data from the available literature.”

I imagine the authors mean: phloem at unusual positions? 

Material & Methods

The name of one author from citation 52 is Quintanar-Castillo & Pace.

Quintanar is misspelled

Results

The first paragraph of results is introduction, doesn’t need to be repeated. 

Also, eliminate “dicots” “higher plants” etc. 

What is DV in Fig. 6? 

Fig. 11, Scientific names are not in italics

I am not very convinced by the successive intraxylary cambia of Ipomoea turbinata. Can the authors put more photos or show in a way that those cells are really cambial initials?  

Line 669. I think it is too strong to state that the formation of radially oriented sieve elements in rays have no taxonomic importance. I don’t think this is true. Once it is found in a taxon, it is typically found there again, so it may be at least an autapomorphy. Either eliminate this statement or easy it down. 

Line 800. It’s Dolichandra unguis-cati, not anguis-cati, correct

Discussion

Line 887 Saying that phloem and xylem are “the most indispensable tissues” is wrong. Entire lineages of plants live without them, such as the Briophytes.

On my opinion, parenchyma is the only indispensable tissue in plants. 

Line 889-890 the idea that evolution occurred FROM pteridophytes TOWARDS Angiosperms as it is posited here is outdated. These are two lineages that evolved independently after the cladogenesis event that gave rise to them. Therefore, ferns are NOT more primitive than Angiosperms. Rephrase.

Also, “pteridophytes”, “dicots”, are not monophyletic groups, and these names were dropped in favor to names that reflect monophyly, such as magnoliids, eudicots, monilophytes, licophytes etc.

Line 924 onwards

Sapindaceae do not have “multiple steles”, because the multiple procambium islands that later give rise to the compound or divided stems are encircled by a single pericycle and endondermis, and therefore it is just ONE stele. 

Line 943. Who is “he” in “he commented”? 

Line 962. Solereder is misspelled 

Line 988. In Nyctaginaceae Cunha Neto et al. showed that for the species with interxylary phloem what is included within the islands is not conjuctive tissue, but phloem (a coalescent phloem parenchyma). Check that and correct here. Therefore, what Carlquist says does apply, but only for instances of successive cambia and not interxylary phloem (even in Caryophyllales).

Line 1105. Campsis do not belong to Apocynaceae, but to Bignoniaceae.

Line 1129. Extra-fascicular cambium is a completely different phenomenon from intraxylary phloem, as per the article that the authors cite. Eliminate or create a new headline for this type of phloem. 

These are my main comments, otherwise I will be very happy to see this article published for its content and quality are really good. I congratulate the authors.

Best wishes,

Marcelo Pace

Author Response

The have some caveats that need addressing, though

The first one regards the title. It doesn't reflect the content of the article, because there isn't anything on phloem regulation, and also not all phloem treated in this article are within xylem. I suggest something like: Inter and intraxylary phloem in plants: subtypes, occurrence, and development. 

or yet: Phloem in odd positions: a review of inter and intraxylary phloem in vascular plants. 

Response: As per suggestion, the title of the manuscript is changed as “Phloem in odd positions: a review of inter and intraxylary phloem in vascular plants”

Also, I urge the authors to eliminate completely citations of the term “dicotyledons” , since it has been dropped decades ago. Substitute for “woody angiosperms” or "magnoliids and eudicots" like they rightfully did in some sections. Do the same throughout the text, whenever “dicots” are mentioned. 

Response: As per suggestion, the term dicotyledons/dicots is replaced with the suggested terms.

Abstract: Make it more clear that this is a review article, because at first I was confused trying to identify what is background and what was the result of this work. Only after reading the entire MS, I understood this is a review paper. Make it clear from start. 

Response: It is now indicated in the type of paper and clarified in the last sentence of the Abstract.

Eliminate “higher plants”, just use “vascular plants” or “euphyllophytes” if wanting to have the lycophytes excluded, or “seed plants” if wanting to have only gymnosperms and angiosperms. The term is used also in other parts of the MS and needs to be eliminated, since it was also dropped long ago.

Response: Changes are made as per suggestion and higher plants is replaced with vascular plants.

I disagree with the following proposition that “The occurrence of such phloem is characteristic to certain families or groups of plants that either have a characteristic habit or habitat.” Intraxylary phloem is ubiquitous in Myrtales, where plants of all habitats and plant habits are to be found. 

Response: The sentence is rephrased and omitted the sentence “groups of plants that either have a characteristic habit or habitat.”

What is “successive xylary phloem cambia”. The term commonly seen in the literature is “successive cambia”. The cambium is neither xylary or phloem, so this would be a misnomer. 

Response: now the term is deleted and thank you for protecting me from the criticism for use of incorrect term. 

Introduction

What are “developmental provocations”? This sentence is an almost identical sentence from the cited paper “Rodríguez-Villalon” only by substituting some words, such as enables for empowers and stimuli for provocation. But provocations read incorrectly here. I suggest completely rephrasing this sentence.

Response: Sentence is rephrased as per suggestion

The authors say sieve cells are a synonym of Strasburger cell. That is a conceptual mistake, since they are completely different cell types. The Strasburger cells are parenchymatic cells associated to the sieve cells, which maintain these cells alive and functioning. 

Response: Term ‘Strasburger cell’ is now deleted. In fact, the sentence related with companion cell in angiosperm and its comparison with other group was removed during editing and by mistake term could not be deleted that conveyed the wrong meaning.

Another conceptual mistake is calling the vascular system of the root a bundle. It’s not a bundle, because it is a protostele, therefore, it is a complete cylinder, and not a collection of bundles.

Response: Sentence is now rephrased.

I disagree that not all contemporary scientists nowadays use phloem as a term. The authors cite a paper from Esau 1939 to support this and the use of other terms such as criblé in French or Cribalheit in German, but these terms are no longer recommended. 

Response:  Dear Sir, I agree with you but in this sentence/para, I have just tried to provide the historical information about phloem, otherwise hardly any textbook talks about this. I have no problem to omit this sentence but I feel history point of view it may be retained.

Eliminate “higher plants” and “lower plants”, these terms have been abolished for decades. Phaeophycean algae are no longer considered plants. 

Response: Both the terms i.e., “higher plants” and “lower plants” are replaced with seed plants (for Gymnosperms and angiosperms) and lycophytes pteridophytes/ferns respectively while term Pheophycean algae is deleted

The citation of Cucurbita in the context of this MS is debatable. Because the authors are talking about intra and interxylary phloem, but the article cited, from Turgeon and Oparka, is actually dealing with a third type of phloem present in the Cucurbitaceae, which they called interfascicular (in between the vascular bundles). It leads the readers to think the paper is talking about the internal phloem (intraxylary phloem). 

Response: Now the sentence is rephrased. Our way of framing the sentence incorrect, in fact it was linked with the next sentence “whether function of sieve elements is decided on its position” but when writing we mixed it with intra-and interxylary phloem. We hope that now it is clear.

The aims are: “The present work aims to compile information on phloem at unusual phloem, their ontogeny and possible role based on our own work by using supporting data from the available literature.”

I imagine the authors mean: phloem at unusual positions? 

Response: yes sir, now it is corrected as “at unusual positions”  

Material & Methods

The name of one author from citation 52 is Quintanar-Castillo & Pace.

Quintanar is misspelled

Response: Corrected as per suggestion “a” was missing now corrected.

Results

The first paragraph of results is introduction, doesn’t need to be repeated. 

Response: Except few lines, major portion of this para is deleted.

Also, eliminate “dicots” “higher plants” etc. 

Response: Both the terms are deleted throughout the text and suggested terms are used.

What is DV in Fig. 6? 

Response: Abbreviations (C = cambium, DV = differentiating vessel) is added in figure legends

Fig. 11, Scientific names are not in italics

Response: All three scientific names italicised

I am not very convinced by the successive intraxylary cambia of Ipomoea turbinata. Can the authors put more photos or show in a way that those cells are really cambial initials?  

Response: To avoid the confusion I have simply rephrased the sentence and removed the word “successive”

Line 669. I think it is too strong to state that the formation of radially oriented sieve elements in rays have no taxonomic importance. I don’t think this is true. Once it is found in a taxon, it is typically found there again, so it may be at least an autapomorphy. Either eliminate this statement or easy it down. 

Response: I have deleted this part (i.e., no taxonomic importance) from the sentence.

Line 800. It’s Dolichandra unguis-cati, not anguis-cati, correct

Response: now, ‘a’ is replaced with ‘u’

Discussion

Line 887 Saying that phloem and xylem are “the most indispensable tissues” is wrong. Entire lineages of plants live without them, such as the Briophytes.

On my opinion, parenchyma is the only indispensable tissue in plants. 

Response: word ‘indispensable’ is replaced with necessary

Line 889-890 the idea that evolution occurred FROM pteridophytes TOWARDS Angiosperms as it is posited here is outdated. These are two lineages that evolved independently after the cladogenesis event that gave rise to them. Therefore, ferns are NOT more primitive than Angiosperms. Rephrase.

Also, “pteridophytes”, “dicots”, are not monophyletic groups, and these names were dropped in favour to names that reflect monophyly, such as magnoliids, eudicots, monilophytes, licophytes etc.

Response: Sentence is rephrased and simplified as per suggestion and omitted peridophytes-gymnosperms and gymnosperms-angiosperms.

Line 924 onwards

Sapindaceae do not have “multiple steles”, because the multiple procambium islands that later give rise to the compound or divided stems are encircled by a single pericycle and endodermis, and therefore it is just ONE stele. 

Response: Sentence is rephrased as ‘presence of compound or divided stem’ and the term ‘multiple stele’ is omitted by.

Line 943. Who is “he” in “he commented”? 

Response: Sentence is now corrected as “The occurrence of interxylary phloem is reported in 21 families by Carlquist [8] and further Carlquist [8, 51] commented that some of them need further study.”

Line 962. Solereder is misspelled 

Response: Spelling is corrected now.

Line 988. In Nyctaginaceae Cunha Neto et al. showed that for the species with interxylary phloem what is included within the islands is not conjunctive tissue, but phloem (a coalescent phloem parenchyma). Check that and correct here. Therefore, what Carlquist says does apply, but only for instances of successive cambia and not interxylary phloem (even in Caryophyllales).

Response: Even I wanted to say the same things that there are two different types. Now I have clearly mentioned about the parenchyma cells.

Line 1105. Campsis do not belong to Apocynaceae, but to Bignoniaceae.

Response: Family name is corrected,

Line 1129. Extra-fascicular cambium is a completely different phenomenon from intraxylary phloem, as per the article that the authors cite. Eliminate or create a new headline for this type of phloem. 

Response: A separate paragraph is created for the comparison between intra-and interxylary phloem and extra fascicular phloem of Cucurbitaceae. Intension behind this para is just to provoke similar studies on inter-and intraxylary phloem. I don’t know how much I have justified inclusion of extra fascicular phloem in the present MS. If needed, it may be deleted since in it is not directly related with present study. This fact has been informed by both the reviewers as well as I have also admitted the fact in the newly phrased para.

I hope that I have justified the queries and if any further changes are suggested, will be incorporated in the further correspondence.

Reviewer 2 Report

The manuscript "Regulation of phloem formation within the secondary xylem of dicotyledons” written by Rajput et al. describes the phenomenon of anomalous phloem formation. In the paper, authors summarized the results of their previous and new research. The strength of the article is the detailed presentation and illustration of atypical formation of secondary phloem, also in the development aspect. However, the manuscript is very long and contains many unnecessary repetitions, therefore, in my opinion, the manuscript as it is, is not prepared well enough for publication and needs the improvement.

The article is correctly structured and very well illustrated.

Main concerns:

 1.      The article is too long and it is not clear whether it is a research paper or a review. It is not known whether the authors refer to new results or to their own results from previous works, which should be cited in detail throughout the text. I would suggest major changes in the manuscript and prepare a review article, but written without structure as for research articles (M&M, Results, Discussion), or write two separate manuscripts: a research paper containing new information about new analysed species, and as the second one - a review article summarizing the knowledge about anomalous forms of secondary phloem formation. However, the presentation of new results would require a very accurate and documented presentation of developmental changes, which is missing in this work. Authors may also consider writing an original article and to shorten the text combine the results with the discussion.

2.      Authors should adapt their article to the requirements contained in the instructions for authors. E.g. abstract is very extensive (well over 200 words) and does not contain the required information.

3.      The title does not correspond to the topic of the work because the manuscript does not mention the regulation of phloem formation.

4.      There are some mistakes in the article - for example (lines 66-67) Strasburger cells are not sieve cells, but they are cells that are found in the vicinity of sieve cells (so-called albumin cells). Sieve cells are also found in gymnosperms, not only in seedless vascular plants. Similarly, between protoxylem and protophloem procambium is located instead of cambium (lines 654-655); Similarly, the Authors should omit “ray cambium” and use “ray cell initials” (line 677), cambium is a tissue consisted of two types of cells ray initials and fusiform initials

5.      There is no anatomical description in the Introduction that would introduce a reader unfamiliar with the subject of anomalous cambium. On the other hand, I would suggest delate a comprehensive historical outline (especially in a research paper). The purpose of the work is also not clearly defined - which may result from the fact that it is not known whether it is the original paper or a review.

6.      In Materials and methods it is not clear whether all plants shown in the table were analysed as described or only the "new" ones marked as "unpublished data".

7.      The results are described more like a discussion, especially since the authors refer to other authors, although they are not cited e.g. lines 247; 476; 843.

8.      Interxylary phloem in Caryophylleaes should be featured with other types of interxylary phloem.

9. The Discussion is  largely a repetition of the results.

10.      I suggest ordering the figures, e.g. 3F should be inserted to Fig 4.

11.   I also found many editorial mistakes at work. E.g.: Figures' captions lack explanations of arrows, arrowheads or letters used in the figures. The abbreviations are in the middle of figure 3 caption, instead of at the end. Sometimes the letters referring to the pictures are bold and sometimes not. Fig 2e and f show the same cells, thus one may be delated.

Author Response

Comments and Suggestions for Authors

  1. The article is too long and it is not clear whether it is a research paper or a review. It is not known whether the authors refer to new results or to their own results from previous works, which should be cited in detail throughout the text. I would suggest major changes in the manuscript and prepare a review article, but written without structure as for research articles (M&M, Results, Discussion), or write two separate manuscripts: a research paper containing new information about new analysed species, and as the second one - a review article summarizing the knowledge about anomalous forms of secondary phloem formation. However, the presentation of new results would require a very accurate and documented presentation of developmental changes, which is missing in this work. Authors may also consider writing an original article and to shorten the text combine the results with the discussion.

Response: Dear Sir/Madam, Present work is authors (KSR and his students) original work published within last decade. During this period, we reported some new types that work is yet to be published which was mentioned as unpublished data and now it is referred as “Present work”. Some of the species are described in general, but unpublished data like Dalechampia (i.e., Figure 7) and Gallesia (i.e., Figure 19) is explained in detail because both the types are not reported so far in the literature. We wanted to write a separate paper on newly analysed species but present MS is the special issue article. Therefore, we thought it is the best platform to represent all different types of inter-and intraxylary phloem reported so far.

As per suggestion, I have tried to shorten the manuscript.

  1. Authors should adapt their article to the requirements contained in the instructions for authors. E.g., abstract is very extensive (well over 200 words) and does not contain the required information.

Response: Now we have tried to shorten the abstract by removing some back ground information about the topic.

  1. The title does not correspond to the topic of the work because the manuscript does not mention the regulation of phloem formation.

Response: As per suggestion, now we have changed the title.

  1. There are some mistakes in the article - for example (lines 66-67) Strasburger cells are not sieve cells, but they are cells that are found in the vicinity of sieve cells (so-called albumin cells). Sieve cells are also found in gymnosperms, not only in seedless vascular plants.

Response: Dear Sir, term ‘Strasburger cell’ is now deleted. In fact, the sentence related with companion cell in angiosperm and its comparison with other group was removed during editing and by mistake term could not be deleted that conveyed the wrong meaning.

Similarly, between protoxylem and protophloem procambium is located instead of cambium (lines 654-655);

Response: Sentence is rephrased, we had mentioned that “as the secondary growth initiates …..” but now the sentence is rephrased as “after the initiation of secondary growth …..

Similarly, the Authors should omit “ray cambium” and use “ray cell initials” (line 677), cambium is a tissue consisted of two types of cells ray initials and fusiform initials

Response: Sentence is corrected as per suggestion and ray cambium is replaced with “ray cell initials”

  1. There is no anatomical description in the Introduction that would introduce a reader unfamiliar with the subject of anomalous cambium. On the other hand, I would suggest delate a comprehensive historical outline (especially in a research paper). The purpose of the work is also not clearly defined - which may result from the fact that it is not known whether it is the original paper or a review.

Response: Dear Sir/Madam, we agree with both the reviewers has asked to remove the historical background of the phloem and we do not mind to delete the same. However, our intension is that there are several developments in the subject but young generation is missing this historical background because most of the recent textbooks talk about current developments in phloem. Therefore, we have briefly recapitulated, yet if suggested we will delete this portion.

  1. In Materials and methods it is not clear whether all plants shown in the table were analyzed as described or only the "new" ones marked as "unpublished data".

Response: Dear Sir/Madam, All the plants mentioned in the present manuscript are personally collected wet preserved, sectioned and studied. Some of them are even communicated for publication. Those which are already published are cited with details so that a reader may refer that article. When it is mentioned as ‘unpublished data’ is under the process i.e., either in writing, submitted for publication or detail analysis (i.e., dimensional details of xylem are to be obtained) is in progress. 

  1. The results are described more like a discussion, especially since the authors refer to other authors, although they are not cited e.g., lines 247; 476; 843.

Response: As per suggestion, we have now rephrased the above said sentences.

  1. Interxylary phloem in Caryophylleaes should be featured with other types of interxylary phloem.

Response: We have omitted the sentence about “Strychnos type”. Possibly we fail to understand the suggestion because it is already under the separate subtitle.

  1. The Discussion is largely a repetition of the results.

Response:

  1. I suggest ordering the figures, e.g., 3F should be inserted to Fig 4.

Response: As per suggestion, Fig. 3F is inserted as Fig. 4A. Since the space for Fig. 3F can not be kept empty; therefore, portion of cambium associated with phloem island is added.

  1. I also found many editorial mistakes at work. E.g.: Figures' captions lack explanations of arrows, arrowheads or letters used in the figures. The abbreviations are in the middle of figure 3 caption, instead of at the end. Sometimes the letters referring to the pictures are bold and sometimes not. Fig 2e and f show the same cells, thus one may be delated.

Response: Now we have gone through the figure captions and corrected/add necessary information.

Now, I hope that you will find our manuscript suitable for its publication in your esteemed journal.

Thanking you with warm regards

Round 2

Reviewer 2 Report

After reading the Authors' responses and the new version of the manuscript, I have the impression that the Authors addressed only minor comments and made only slight corrections, while the article should have been significantly rephrased. The arrangement of the manuscript chapters suggests that the article is still the original, almost unchanged version, therefore the description in the text still raises doubts about the type of article it is supposed to be.  Unfortunately, adding one sentence at the end of the abstract, indicating that it is a review, in my opinion is not enough. Therefore, in conclusion, the article has not been rephrased and improved in such a sufficient and proper way to take the form of a review and in this form it is still not suitable for publication.

The comment to shorten the article was not taken into consideration similarly as the remark about necessity of adaptation of editorial requirements in the manuscript (the abstract is still too long).

The comment „There is no anatomical description in the Introduction…” was also omitted. The Authors did not include this suggestion into the text, nor did they address it in their responses to the reviewer’s comments.

There is also no response to the remark „The discussion is largely a repetition of the results”, which, together with earlier suggestions of the reviewer indicated the necessity to rewrite the article as a whole, and not just make minor corrections.  Especially that in the review article there is no part: ‘discussion’ as the whole manuscript is a discussion demanding from the authors a well-prepared and critical review of findings.

Also other corrections, requiring more time-consuming changes and modifications (e.g. removing one of the two photos Fig 2e or f) were not done. The lack of these changes, unfortunately, means that the Authors did not refer to the main and serious comments of the reviewer.

Author Response

Response: Dear sir/madam, we do not have any intension to disobey the suggestion, we were reluctant to omit the information, which is totally removed from the text for the sake of maintaining 200 words in the Abstract. Now, the background information given in the abstract is almost removed and we feel that it was important for general readers.

  • The comment “There is no anatomical description in the Introduction…” was also omitted. The Authors did not include this suggestion into the text, nor did they address it in their responses to the reviewer’s comments.

Response: Now, we have added some information which was missing in the earlier version. In the earlier version, structure and composition of phloem was already mentioned along with the flow of sentences that were concentrating on structure, function, ontogeny, etiology of the terms while describing the terms.

In fact, it was not clear to us that what kind of anatomical description is to be added? Even, at present, we have doubt whether same information was expected or not during this revision. We hope the esteemed reviewer will agree with the present version.

  • There is also no response to the remark, the discussion is largely a repetition of the results”, which, together with earlier suggestions of the reviewer indicated the necessity to rewrite the article as a whole, and not just make minor corrections.  Especially that in the review article there is no part: ‘discussion’ as the whole manuscript is a discussion demanding from the authors a well-prepared and critical review of findings.

Response: Now, we have completely omitted the discussion and clubbed the results and discussion. As per suggestion, all subtitle (introduction, Materials and methods, Results and Discussion) is removed and tried to omit the repetition.

  • Also, other corrections, requiring more time-consuming changes and modifications (e.g., removing one of the two photos Fig 2e or f) were not done. The lack of these changes, unfortunately, means that the Authors did not refer to the main and serious comments of the reviewer.

Response: Dear Sir/madam, in earlier version, Figure 3F was asked to insert at 4A position, that we have already done but this time Figure 2E or 2F is suggested. We feel that both the images (2E or 2F) are necessary to distinguish that the images provided are from xylem only and not from the phloem. Enlarged view (2F) clearly shows 2 companion cells with each sieve tube element.